**Present-Day Methane Shortwave Absorption Mutes Surface Warming Relative to Preindustrial Conditions**

Robert J. Allen[1], Xueying Zhao[1,2,3], Cynthia A. Randles[4*], Ryan J. Kramer[5], Bjørn H. Samset[6] and Christopher J. Smith[7,8]

[1]Department of Earth and Planetary Sciences, University of California, Riverside, CA, USA.

[2]National Center for Atmospheric Research, Boulder, CO.

[3]Department of Earth and Planetary Science, The University of Texas at Austin, Austin, TX, USA.

[4]ExxonMobil Technology and Engineering Company, Annandale, NJ, USA
[*]now at UNEP International Methane Emission Observatory, Paris, France.

[5]NOAA Geophysical Fluid Dynamics Laboratory, Princeton, NJ.

[6]CICERO Center for International Climate and Environmental Research in Oslo, Oslo, Norway.

[7]School of Earth and Environment, University of Leeds, Leeds, UK.

[8]International Institute for Applied Systems Analysis (IIASA), Laxenburg, Austria.

*Correspondence to*: Robert J. Allen (rjallen@ucr.edu)

**Short Summary:**

Present-day methane shortwave absorption mutes 28% (7-55%) of the surface warming associated with its longwave absorption. The precipitation increase associated with the longwave radiative effects of the present-day methane perturbation is also muted by shortwave absorption but not significantly so. Methane shortwave absorption also impacts the magnitude of its climate feedback parameter, largely through the cloud feedback.

**Abstract**. Recent analyses show the importance of methane shortwave absorption,
which many climate models lack. In particular, Allen et al. (2023) used idealized
climate model simulations to show that methane shortwave absorption mutes up to
30% of the surface warming and 60% of the precipitation increase associated with
its longwave radiative effects. Here, we explicitly quantify the radiative and
climate impacts due to shortwave absorption of the present-day methane
perturbation. Our results corroborate that present-day methane shortwave
absorption mutes the warming effects of longwave absorption. For example, the
global mean cooling in response to the present-day methane shortwave absorption
is $-0.10 \pm 0.07$ K, which offsets 28% (7-55%) of the surface warming associated
with present-day methane longwave radiative effects. The precipitation increase
associated with the longwave radiative effects of the present-day methane
perturbation ($0.012 \pm 0.006$ mm d$^{-1}$) is also muted by shortwave absorption but
not significantly so ($-0.008 \pm 0.009$ mm d$^{-1}$). The unique responses to methane
shortwave absorption are related to its negative top-of-the-atmosphere effective
radiative forcing but positive atmospheric heating, and in part methane's
distinctive vertical atmospheric solar heating profile. We also find that the present-
day methane shortwave radiative effects, relative to its longwave radiative effects,
are about five times larger than those under idealized carbon dioxide perturbations.
Additional analyses show consistent but non-significant differences between the
longwave versus shortwave radiative effects for both methane and carbon dioxide,
including a stronger (negative) climate feedback when shortwave radiative effects
are included (particularly for methane). We conclude by reiterating that methane
remains a potent greenhouse gas.

## 1 Introduction

Several recent studies (Li et al., 2010; Etminan et al., 2016; Collins et al., 2018; Byrom and Shine, 2022) have shown the significance of methane ($CH_4$) shortwave (SW) absorption—which is lacking in many climate models (Forster et al., 2021)—at near-infrared (NIR) wavelengths. Etminan et al. (2016) first showed methane SW absorption increases its stratospherically adjusted radiative forcing (SARF) by up to ~15% as compared to its longwave (LW) SARF. Smith et al. (2018) subsequently inferred negative rapid adjustments (i.e., surface temperature independent responses; see Section 2) due to $CH_4$ SW absorption, using four of ten models from the Precipitation Driver and Response Model Intercomparison Project (PDRMIP; Myhre et al., 2017) that included an explicit representation of methane SW absorption. Byrom and Shine (2022) showed that $CH_4$ SW forcing depends on several factors, including the spectral variation of surface albedo, the vertical profile of methane, and absorption of solar radiation at longer wavelengths, specifically methane's 7.6 µm band. They estimated a smaller impact of $CH_4$ SW absorption, with a 7% increase in SARF, in part due to the inclusion of the 7.6 µm band which mainly impacts stratospheric solar absorption.

The recent analysis of Allen et al. (2023) (hereafter referred to as A23) used Community Earth System Model version 2 (CESM2; Danabasoglu et al., 2020) simulations to isolate the effects of $CH_4$ SW absorption, and showed that it muted the surface warming and wetting due to methane's LW radiative effects. Muting of surface warming was attributed largely to cloud rapid adjustments, including increased low-level clouds and decreased high-level clouds. These cloud changes in turn were associated with the vertical profile of atmospheric solar heating, and corresponding changes to atmospheric temperature and relative humidity.

We adopt similar terminology as in A23. Throughout this manuscript, the terms "SW radiative effect"/"SW absorption" and "LW radiative effect" refers to the radiative effects of methane (and eventually carbon dioxide) on the climate system as isolated by a suite of simulations (to be discussed below). This terminology is used interchangeably with the abbreviations "$CH_{4SW}$" and "$CH_{4LW}$", respectively.

A23 focused on three idealized methane perturbations, including 2x, 5x and 10x preindustrial methane concentrations. Relatively large perturbations were emphasized to maximize the signal to noise ratio, as well as to robustly identify mechanisms. Despite these relatively large methane perturbations, 5x preindustrial methane concentrations are comparable to end of 21$^{st}$ century projections under the Shared Socioeconomic Pathway 3-7.0 (i.e., 0.75 ppm to 3.4 ppm). Although

5xCH$_4$ and 10xCH$_4$ SW radiative effects showed a clear muting of the
corresponding LW effects, 2xCH$_4$ did not.  For example, the global mean near-
surface air temperature (TAS) response under 5xCH$_{4SW}$ and 10xCH$_{4SW}$ yielded
significant global cooling at $-0.23 \pm 0.07$ and $-0.39 \pm 0.07$ K.  We reiterate that
this cooling is due to isolation of methane shortwave absorption alone; the total
(including methane's longwave absorption) temperature response is significant
warming at $0.45 \pm 0.05$ and $0.85 \pm 0.05$ K, respectively (i.e., longwave
absorption effects dominate). 2xCH$_{4SW}$, however, yielded a warming response of
$0.06 \pm 0.06$ K that is not significant at the 90% confidence level.  Similar results
apply for the global mean precipitation (P) response, where a significant decrease
occurred under 5xCH$_{4SW}$ and 10xCH$_{4SW}$ at $-0.021 \pm 0.008$ and $-0.039 \pm$
$0.008$ mm d$^{-1}$ (-0.7 and -1.3%).  For 2xCH$_{4SW}$, the response was again not
significant at $0.002 \pm 0.008$ mm d$^{-1}$ (0.06%).  The lack of significant climate
responses in the 2xCH$_{4SW}$ coupled ocean-atmosphere simulation is consistent with
its relatively weak forcing as compared to the larger methane perturbations, and
relative to internal climate variability of the coupled ocean-atmosphere system.
Here we conduct analogous simulations as A23 to explicitly calculate the
shortwave absorption effects of the present-day methane concentration, i.e., the
~750 to ~1900 ppb increase (~2.5x).  Our results support the prior conclusions
from A23.  We further expand upon our understanding of the climate effects of
CH$_{4SW}$ by conducting an atmospheric energy budget analysis and by evaluating the
climate feedback and hydrological sensitivity parameters (and climate sensitivity),
and by comparing the effects of methane SW absorption with those from carbon
dioxide SW absorption.

**2 Materials and Methods**

An array of targeted methane-only and carbon dioxide-only equilibrium time slice
(i.e., cyclic repetition of the imposed perturbation) climate simulations are
conducted with CESM2 (Danabasoglu et al., 2020), which includes the most recent
model components such as the Community Atmosphere Model version 6 (CAM6).
CAM6's radiation parameterization, the Rapid Radiative Transfer Model for
general circulation models (RRTMG; Iacono et al., 2008) includes a representation
of CH$_4$ SW absorption in three near-infrared bands including 1.6-1.9 μm, 2.15-2.50
μm and 3.10-3.85 μm.  Methane shortwave absorption at 7.6 μm (the mid-infrared;
mid-IR), however, is not represented.  Furthermore, although CESM2 includes a
representation of CH$_4$ SW absorption, RRTMG underestimates CH$_4$ (and CO$_2$) SW
IRF by 25-45% (Hogan and Matricardi, 2020).
Our focus here is a set of 2.5x preindustrial atmospheric $CH_4$ concentration
simulations, to complement the three methane perturbations (2x, 5x and 10x
preindustrial atmospheric $CH_4$ concentrations) performed by A23. We perform
both fixed climatological sea surface temperatures (fSST) and fully coupled ocean-
atmosphere simulations (Table 1), and conduct two sets of identical experiments,
one that includes $CH_4$ LW+SW radiative effects ($2.5xCH_4^{EXP}$) and one that lacks
$CH_4$ SW radiative effects ($2.5xCH_{4NOSW}^{EXP}$). $CH_4$ SW absorption in the three NIR
bands in RRTMG is turned off in the simulations that lack methane SW
absorption. These are compared to a default preindustrial control experiment
($PIC^{EXP}$), which includes $CH_4$ (as well as other radiative species such as $CO_2$)
LW+SW radiative effects, as well as to a preindustrial control experiment with
$CH_4$ SW radiative effects turned off (i.e., LW effects only, denoted as
$PIC_{NOCH4SW}^{EXP}$). To clarify, SW changes can still be present in $2.5xCH_{4NOSW}^{EXP}$, but
only as a rapid adjustment (or a temperature-induced response) associated with the
direct LW absorption of methane. For example, direct LW absorption of methane
can drive changes in water vapor and clouds, which in turn could impact SW
radiation.
This suite of $CH_4$ simulations allows quantification of the $CH_4$ LW+SW, LW and
SW radiative effects, denoted as 2.5x$CH_{4LW+SW}$, 2.5x$CH_{4LW}$ and 2.5x$CH_{4SW}$. The
2.5x$CH_{4LW+SW}$ signal is obtained by subtracting the default 2.5x$CH_4$ perturbation
from the default control ($2.5xCH_4^{EXP} - PIC^{EXP}$). The 2.5x$CH_{4LW}$ signal is
obtained by subtracting the 2.5x$CH_4$ perturbation without $CH_4$ SW absorption from
the corresponding control simulation without $CH_4$ SW absorption
($2.5xCH_{4NOSW}^{EXP} - PIC_{NOCH4SW}^{EXP}$). The 2.5x$CH_{4SW}$ signal is obtained by taking the
double difference, i.e., ($2.5xCH_4^{EXP} - PIC^{EXP}) - (2.5xCH_{4NOSW}^{EXP} - $
$PIC_{NOCH4SW}^{EXP}$). The 2.5x$CH_{4SW}$ signal therefore represents $CH_4$ SW absorption and
also the impacts of this SW absorption on $CH_4$ LW rapid adjustments (and surface
temperature responses). We also calculate the corresponding instantaneous
radiative forcing (IRF), which is defined as the initial perturbation to the radiation
balance, using the Parallel Offline Radiative Transfer (PORT) model (Conley et
al., 2013). PORT isolates the RRTMG radiative transfer computation from the
CESM2-CAM6 model configuration.
Fixed SST experiments are used to estimate the 'fast' climate responses and the
effective radiative forcing (ERF). ERF is defined as the top-of-the-atmosphere
(TOA) net radiative flux difference between the experiment and control simulation,
with climatological fixed SSTs and sea-ice distributions without any adjustments
for changes in the surface temperature over land (Forster et al., 2016). ERF can be
decomposed into the sum of the IRF and rapid adjustments (ADJs). Rapid
adjustments represent the change in state in response to the initial perturbation (i.e.,
IRF) excluding any responses related to changes in sea surface temperatures. Rapid
adjustments, which for example include clouds and water vapor, are estimated
using the radiative kernel method (Soden et al., 2008; Smith et al., 2018, 2020)
applied to the climatological fixed SST simulations.  A radiative kernel is basically
the partial derivative of the radiative flux with respect to a variable (e.g., moisture)
that changes with temperature.  It therefore represents the radiative impacts from
small perturbations in a state.  To calculate the rapid adjustments, the radiative
kernel is multiplied by the change in the climate variable under consideration
(from the fSST simulations). The Python-based radiative kernel toolkit of Soden et
al. (2008), along with the Geophysical Fluid Dynamics Laboratory radiative
kernel, are used here.  The method for calculating cloud rapid adjustments with
radiative kernels is a bit more involved.  Here, we use the kernel difference method
(Smith et al., 2018) which employs a cloud-masking correction applied to the
cloud radiative-forcing diagnostics.  The cloud-masking correction is based on the
kernel-derived non-cloud adjustments and IRF.  A23 showed that this methodology
performed well, including a small residual term (i.e., $ERF - IRF - \Sigma ADJs <$
$\sim 5\%$ of ERF).  Furthermore, similar results were obtained with an alternative
radiative kernel based on CloudSat/CALIPSO (Kramer et al., 2019).
The total climate response, which includes the IRF, ADJs and the surface
temperature responses, is quantified using the coupled ocean-atmosphere
experiments.  Specifically, the radiative effects associated with the total climate
response are estimated using the same radiative kernel decomposition as above, but
applied to the coupled ocean-atmosphere simulation.  The surface temperature
responses (i.e., 'slow' response) are estimated as the difference between the
coupled ocean atmosphere simulations and the climatologically fixed SST
experiments.  Similarly, the radiative effects associated with the slow response are
calculated as the difference between the kernel-derived radiative effects of the total
and fast responses.
To reiterate, our framework is to decompose the total response (directly estimated
from coupled simulations) into a fast (surface temperature independent) response
and a slow (surface temperature dependent) response:

221              Total Response = Fast Response + Slow Response      (1)

The fast response is directly estimated from the fSST simulations and includes the
rapid adjustments.  The slow response is estimated from the difference of the total
and fast responses (i.e., coupled simulation minus fSST simulation).  This is
consistent with the IPCC framework, which uses the concepts of an adjustment to
an imposed forcing (i.e., independent of surface temperature) and a radiative
response to a global mean temperature change. It is also analogous to the
methodology employed in several other papers, including many PDRMIP papers
(e.g., Samset et al., 2016; Myhre et al., 2017).
Our simulations are performed at 1.9° x 2.5° latitude-longitude resolution with 32
atmospheric levels. Coupled ocean-atmosphere experiments are initialized from a
spun-up preindustrial control simulation and subsequently integrated for 90 years.
Total climate responses are estimated using the last 40 years of these coupled
ocean-atmosphere experiments. As climatologically fixed SST simulations
equilibrate more quickly, these are run for 32 years. The ERF and rapid
adjustments are estimated from the last 30 years of these fSST experiments.
Our integration lengths are consistent with other related idealized time-slice studies
including for example a 100-year integration (and analysis of the last 50 years) of
coupled simulations under PDRMIP (e.g., Samset et al., 2016; Myhre et al., 2017).
A similar statement applies for the integration length of our fSST runs, e.g., the
Radiative Forcing Model Intercomparison Project (RFMIP; Pincus et al., 2016)
specifies 30-year fSST simulations.
We note that even with a 90-year coupled ocean simulation, the model has not yet
reached equilibrium. Given computational resource limitations, there is always a
tradeoff between the number of simulations performed and length of each
simulation.

A two-tailed pooled $t$ test is used to assess the statistical significance of a climate
response, based on the annual mean difference between the experiment and
control. We evaluate a null hypothesis of zero difference with $n_1 + n_2 - 2$ degrees
of freedom. Here, $n_1$ and $n_2$ are the number of years in the experiment and control
simulations (e.g., 40 years for the coupled ocean-atmosphere runs). The pooled
variance $S_p^2 = \frac{(n1-1)S_1^2 + (n2-1)S_2^2}{n1+n2-2}$ is used, where $S_1^2$ and $S_2^2$ are the sample variances.
Quoted uncertainty estimates are based on the 90% confidence interval using the
pooled variance according to 1.65*$S_p$.

**3 Results**
**3.1 2.5xCH$_4$ Radiative Flux Components & Rapid Adjustments**
Figure 1a shows the 2.5xCH$_4$ TOA ERF, IRF and ADJ, as well as the radiative
kernel decomposition of ADJ (Fig. 1b). The 2.5xCH$_4$ TOA LW IRF is $0.46 \pm$
$0.05$ W m$^{-2}$ and the corresponding TOA SW IRF is $0.06 \pm 0.07$ W m$^{-2}$ (not
significant at the 90% confidence level).
The 2.5xCH$_4$ instantaneous shortwave heating rate (QRS) profile (Figure 2a)
exhibits positive values for atmospheric pressure levels less than ~700 hPa and
negative values for pressure levels greater than ~700 hPa. As discussed in A23,
increasing the atmospheric methane concentration does not increase lower-
tropospheric SW heating because the three near-infrared bands are already highly
saturated here (e.g., due to water vapor absorption). Furthermore, the methane-
induced QRS increase aloft decreases the available solar radiation in the three
near-IR methane absorption bands (1.6-1.9 μm, 2.15-2.50 μm and 3.10-3.85 μm)
that can be absorbed by other gases (e.g., water vapor) in the lower-troposphere.
This results in the decrease in SW heating-rate in the lower troposphere (Fig. 2a).
Both of these features exist under 2.5xCH$_{4SW}$ and are consistent with the other
methane perturbations, with the larger perturbations (e.g., 5xCH$_{4SW}$), yielding
larger QRS increases aloft and larger QRS decreases in the lower troposphere.
As mentioned above, A23 showed that methane SW radiative effects lead to a
negative rapid adjustment (largely due to changes in clouds) that acts to cool the
climate system. A positive ADJ represents a net energy increase, whereas a
negative ADJ represents a net energy decrease. Individual rapid adjustments, as
well as the total adjustment, under 2.5xCH$_4$ are displayed in Figure 1b. Under
2.5xCH$_{4SW}$, the total rapid adjustment is $-0.16 \pm 0.10$ W m$^{-2}$, which is largely due
to the cloud adjustment at $-0.12 \pm 0.08$ W m$^{-2}$. The stratospheric temperature
adjustment contributes the remainder at $-0.04 \pm 0.01$ W m$^{-2}$. The remaining terms
(i.e., surface temperature, tropospheric temperature, surface albedo and water
vapor adjustments), most of which are not significant at the 90% confidence level,
have a net zero contribution to the total adjustment (i.e., their sum is zero). Thus,
similar to the larger CH$_4$ perturbations in A23, 2.5xCH$_{4SW}$ yields a significant
negative total rapid adjustment that is largely due to the cloud adjustment.
This negative rapid adjustment promotes a negative ERF under methane SW
absorption. We reiterate that the negative ERF is due to isolation of methane
shortwave absorption alone; methane's longwave effects still dominate the ERF.
This is because the ERF is the sum of ADJs and IRF. For example, under the larger
5xCH$_{4SW}$ perturbation in A23, the ERF and ADJ were both significant at $-0.22 \pm$
$0.17$ W m$^{-2}$ and $-0.36 \pm 0.13$, respectively. Under 2.5xCH$_{4SW}$, the ERF and ADJ
(Fig. 1a) are $-0.10 \pm 0.13$ W m$^{-2}$ and $-0.16 \pm 0.10$ W m$^{-2}$, respectively, with the
latter significant at the 90% confidence level.  As with the larger methane
perturbations, 2.5xCH$_{4SW}$ offsets (although not significantly so) ~20% of the ERF
associated with 2.5xCH$_{4LW}$ ($0.53 \pm 0.11$ W m$^{-2}$).
The corresponding surface CH$_{4SW}$ "ERFs" (not shown) are more negative than
those at the TOA, at $-0.18 \pm 0.10$ W m$^{-2}$ for 2.5xCH$_{4SW}$ (significant at the 95%
confidence interval).  We note that technically this is not an ERF, but we retain this
terminology since it is calculated analogously to ERF, just using surface as
opposed to TOA radiative fluxes.  This negative surface ERF is consistent with
negative surface CH$_{4SW}$ IRF values (due to atmospheric solar absorption, which
decreases surface solar radiation), and the vertical redistribution of shortwave
heating (Fig. 2a) that drives a negative surface rapid adjustment that is again
largely due to the cloud adjustment.  The surface 2.5xCH$_{4SW}$ IRF value is $-0.10 \pm$
$0.05$ W m$^{-2}$ and the corresponding sum of the surface rapid adjustments is
$-0.08 \pm 0.07$ W m$^{-2}$ (not shown).

## 3.2 2.5xCH$_{4SW}$ Fast Climate Response

Figure 2b-f shows global mean vertical response profiles from the fSST
simulations for the four methane shortwave absorption perturbations (e.g.,
2.5xCH$_{4SW}$). 2.5xCH$_{4SW}$ yields QRS increases (Fig. 2b) in the upper
troposphere/lower stratosphere, as well as QRS decreases in the lower-troposphere.
This is consistent with the aforementioned instantaneous QRS profile response
(Fig. 2a). These changes are associated with temperature (Fig. 2c) and relative
humidity (RH; Fig. 2d) changes that favor increases in low-level cloud cover
(CLOUD; Fig. 2e) that peak near 800 hPa and decreases in high-level cloud cover
(e.g., for pressures < 300 hPa).  Both of these CLOUD responses act to cool the
surface. These cloud changes become larger under the larger methane
perturbations.  For example, 2.5xCH$_{4SW}$ yields a decrease in global mean lower-
tropospheric (pressures > 800 hPa) temperature of $-0.02 \pm 0.02$ K (not significant
at the 90% confidence level) and an increase in upper-tropospheric (between 100
and 500 hPa) temperature of $0.09 \pm 0.04$ K (significant at the 95% confidence
level).  Similarly, global mean lower-tropospheric RH increases by $0.01 \pm 0.06$ %
and upper-tropospheric RH decreases by $-0.09 \pm 0.10$ % (however, both changes
are not significant at the 90% confidence level).  Global mean lower-tropospheric
CLOUD increases by $0.045 \pm 0.04$ % (low cloud as quantified in CESM2 yields
$0.08 \pm 0.07$%; Supplementary Table 1) and upper-tropospheric CLOUD decreases
by $-0.07 \pm 0.04$ %.
Correlations between the 2.5xCH$_{4SW}$ global mean vertical response profiles are
significant.  For example, the correlation between the global mean vertical
temperature and QRS response profile from 990 hPa to 100 hPa is 0.93. The
corresponding correlation between temperature and RH is -0.89, and the
corresponding correlation between RH and CLOUD is 0.80.  Thus, an increase in
SW heating is associated with warming whereas a decrease in SW heating is
associated with cooling.  Warming is associated with a decrease in RH whereas
cooling is associated with an increase in RH.  Furthermore, an increase in RH is
associated with an increase in CLOUD whereas a decrease in RH is associated
with a decrease in CLOUD.  These results help to support the importance of
atmospheric SW absorption in driving the CLOUD response through altered
temperature and RH.  Spatial correlations at specific pressure levels also yield
similarly significant but somewhat weaker correlations (Supplementary Figure 1).
For example, spatially correlating the global mean annual mean change in CLOUD
with the corresponding change in RH yields significant correlations in the lower-
troposphere ranging from 0.40 to 0.65, as well as in the upper-troposphere ranging
from 0.71 to 0.81.  Similar conclusions are obtained with the larger methane
perturbations.
These cloud changes are similar to those that occur in response to absorbing
aerosols like black carbon (i.e., the aerosol-cloud semi-direct effect; Amiri-
Farahani et al., 2019; Allen et al., 2019).  Black carbon solar heating warms and
dries (decreased relative humidity) the free troposphere, which promotes less cloud
cover in the mid- to upper-troposphere (Stjern et al., 2017). Warming aloft (and
cooling of the lower troposphere under CH$_{4SW}$) also suggest enhanced lower-
tropospheric stability.  As lower-tropospheric stability is a measure of the inversion
strength that caps the boundary layer, enhanced lower-tropospheric stability traps
more moisture in the marine boundary layer, allowing for enhanced cloud cover
(e.g., Wood and Bretherton, 2006). Under 2.5xCH$_{4SW}$, global mean lower-
tropospheric stability (estimated here as the temperature difference between 600
hPa and 990 hPa) significantly increases (at the 95% confidence level) by $0.03 \pm$
0.02 K.  Larger increases in lower-tropospheric stability occur under the larger
methane perturbation, e.g., $0.06 \pm 0.02$ K under 10xCH$_{4SW}$ (and similarly, larger
increases in low clouds occur at $0.36 \pm 0.10\%$; Supplementary Table 1).  This
increase in low cloud cover, most of which occurs over the oceans (Supplementary
Figure 2a,d,g,j), is consistent with the increase in lower-tropospheric stability.
Furthermore, enhanced stability also suggests reduced convective mass flux in the
mid/upper-troposphere.  Although we did not archive convective mass flux, Fig. 2f
shows changes in convective cloud cover (CONCLOUD).  All methane
perturbations show decreased CONCLOUD in the mid/upper troposphere
(pressures < 800 hPa). CONCLOUD also increases in the lower-troposphere
(peaking near 900 hPa). Although these CONCLOUD changes are weaker than
those associated with CLOUD, their profiles are very similar, implying that
changes in convection also contribute to changes in CLOUD.
**3.3 2.5xCH$_{4SW}$ Total Climate Response**
Figure 3a-e shows global mean vertical total climate response profiles from the
coupled ocean-atmosphere simulations for the four methane shortwave absorption
perturbations (e.g., 2.5xCH$_{4SW}$). The QRS, RH and CLOUD responses are similar
to those from the fSST simulation (Fig. 2), which further highlights the importance
of rapid adjustments to the total climate response. For example, similar to the fast
response, the total response features increases in low- and mid-level clouds (Fig.
3c; peaking near 800 hPa) and decreases in high-level clouds (for pressures < 300
hPa) occurs, both of which act to cool the surface (Fig. 3f).
Relative to the fast responses discussed above, the total responses are generally
similar but larger and more significant in the lower (and mid) troposphere but
weaker in the upper troposphere. This is consistent with allowing the surface to
respond to the CH$_{4SW}$ perturbation in the fully coupled ocean-atmosphere
experiments, and in particular, the negative surface CH$_{4SW}$ "ERFs" discussed in
Section 3.1 (i.e., decrease in surface solar radiation). For example, the 2.5xCH$_{4SW}$
total response features a decrease in global mean lower-tropospheric temperature
(Fig. 3b) of $-0.10 \pm 0.07$ K which is significant at the 95% confidence level and
about 5x as large as the cooling under the fast response (Fig. 2c). This smaller
lower-tropospheric temperature adjustment (i.e., fast response) is consistent with
the experimental design (i.e., fixed SSTs). A non-significant decrease in upper-
tropospheric temperature of $-0.02 \pm 0.11$ K occurs under the total response, in
contrast to the upper-tropospheric warming under the fast response (Fig. 2c).
Similarly, global mean lower-tropospheric RH (Fig. 3d) increases by $0.05 \pm 0.05$
% (significant at the 90% confidence level) under the 2.5xCH$_{4SW}$ total response,
with a non-significant change in upper-tropospheric RH of $-0.02 \pm 0.08$ %.
Global mean lower-tropospheric CLOUD (Fig. 3c) increases by $0.12 \pm 0.07$ %
(significant at the 99% confidence level) and upper-tropospheric CLOUD
decreases by $-0.06 \pm 0.03$ % (significant at the 99% confidence level). The
corresponding changes under the fast response (Fig. 2) are generally similar, but
smaller in the lower-troposphere (i.e., smaller increases in RH and CLOUD) but
larger in the upper-troposphere (i.e., larger decreases in RH and CLOUD). The
total response of CONCLOUD (Fig. 3e) is generally similar to the fast response
(Fig. 2f), although the 2.5xCH$_{4SW}$ total response lacks an increase in the lower-
troposphere.

Global maps of the TAS and P total climate responses (from coupled ocean-
atmosphere simulations) under 2.5xCH$_{4SW}$ are shown in Fig. 3f,g. The global mean
TAS response is $-0.10 \pm 0.07$ K (significant at the 95% confidence level); the
global mean P response is $-0.008 \pm 0.009$ mm d$^{-1}$ (-0.27%) which is not
significant at the 90% confidence level. Comparing these 2.5xCH$_{4SW}$ responses to
the corresponding 2.5xCH$_{4LW}$ responses of $0.36 \pm 0.05$ K and $0.012 \pm 0.006$ mm
d$^{-1}$ shows that under 2.5xCH$_4$, methane shortwave absorption offsets 28% (7-55%)
of the surface warming and 66% of the precipitation increase associated with its
longwave radiative effects. Although the 66% muting of the precipitation increase
is not significant, this percentage is qualitatively consistent with the larger methane
perturbations.

As noted in Section 3.1, consistent with the larger methane perturbations, the
2.5xCH$_{4SW}$ ERF at $0.10 \pm 0.13$ W m$^{-2}$ offsets 19% (although not significant) of the
ERF associated with 2.5xCH$_{4LW}$. In contrast, 2.5xCH$_{4SW}$ offsets a larger
percentage of the surface warming associated with 2.5xCH$_{4LW}$ at 28%. Based on
the global mean TOA energy decomposition equation $\Delta N = \Delta F + \alpha \Delta TAS$ (e.g.,
Forster et al., 2021), where $\Delta N$ is the change in the global mean TOA net energy
flux [W m$^{-2}$]; $\Delta TAS$ is the change in global mean near-surface air temperature [K];
$\Delta F$ is the change in the global mean TOA net energy flux [W m$^{-2}$] when $\Delta TAS = 0$
(i.e., the effective radiative forcing, ERF); and $\alpha$ is the net feedback parameter [W
m$^{-2}$ K$^{-1}$], if $\Delta F$ is reduced by X%, $\Delta TAS$ should also be reduced by X% assuming a
constant $\alpha$. Supplementary Table 2 and Supplementary Figure 3 show the
individual components of the TOA energy decomposition equation, including the
estimated climate feedback parameter (details on how these are calculated are
included in the corresponding captions). The climate feedback parameter is always
larger (in magnitude) under the various SW+LW signals (e.g., 2.5xCH$_{4LW+SW}$) as
compared to the LW-only signal (e.g., 2.5xCH$_{4LW}$), which suggests the climate
system does not have to warm as much to offset the same TOA energy imbalance
when SW effects are included. However, $\alpha$ has a relatively large uncertainty and it
is not significantly different between the various SW+LW signals and the
corresponding LW-only signals. For example, the climate feedback parameter is
$-1.80 \pm 0.44$ W m$^{-2}$ K$^{-1}$ for 10xCH$_{4LW+SW}$ and $-1.45 \pm 0.26$ W m$^{-2}$ K$^{-1}$ for
10xCH$_{4LW}$. The SW signal consistently (outside of 2.5xCH$_{4SW}$) yields the smallest
(negative) $\alpha$. The corresponding value for 10xCH$_{4SW}$ is $-0.73 \pm 1.08$ W m$^{-2}$ K$^{-1}$.
We also note that the 2.5xCH$_{4SW}$ $\alpha$ has an unphysical positive value (but again
with large uncertainty) at $0.87 \pm 3.41$ W m$^{-2}$ K$^{-1}$. Thus, the climate feedback
parameter is not significantly different under the LW-only effects versus SW
effects of CH$_4$. This uncertainty also helps to explain why the SW effect
contributes different percentages (which are not significant under 2.5xCH$_4$) for
ERF and $\Delta$TAS. Additional analyses (Section 3.7), however, show that there are
significant differences in the cloud feedback (largely due to low clouds) that lend
additional support to the notion that the climate feedback parameter is different
(less negative) under methane SW radiative effects.
Analogous conclusions exist for the climate sensitivity parameter $\lambda$ (K [W m$^{-2}$]$^{-1}$;
i.e., $-1 \times \alpha^{-1}$ ). $\lambda$ is consistently smaller under the various SW+LW signals
relative to the corresponding LW-only signals (Supplementary Table 2), implying
less warming in response to the same TOA energy imbalance when SW effects are
included. The SW signal (outside of 2.5xCH$_{4SW}$) consistently yields the largest $\lambda$,
implying relatively large temperature change in response to the same TOA energy
imbalance. Again, however, the uncertainty is large and these differences are not
significant. For example, the climate sensitivity parameter is $0.55 \pm 0.13$ K [W m$^{-2}$]$^{-1}$ under 10xCH$_{4LW+SW}$ versus $0.69 \pm 0.12$ K [W m$^{-2}$]$^{-1}$ under 10xCH$_{4LW}$. The

corresponding $\lambda$ under 10xCH$_{4SW}$ is $1.37 \pm 2.02$ K [W m$^{-2}$]$^{-1}$.

**3.4 2.5xCH$_{4SW}$ Slow Climate Response**
We apply the radiative kernel decomposition to the 2.5xCH$_{4SW}$ coupled ocean-
atmosphere simulation (Figure 4; Supplementary Figure 4 shows the corresponding
results for 2.5xCH$_{4SW+LW}$ and 2.5xCH$_{4LW}$). The 'fast' responses from the fixed
climatological SST runs (i.e., the rapid adjustments) and the surface-temperature-
induced 'slow' responses (i.e., the difference between the coupled ocean
atmosphere and fixed climatological SST simulations) are also included. Here, a
positive slow response has the same meaning as a positive fast response (ADJ), as
both represent a net energy increase. Similarly, a negative slow response has the
same meaning as a negative ADJ, as both represent a net energy decrease (i.e., we
do not normalize by the change in surface air temperature as is done to calculate a
climate feedback). As with the larger methane perturbations, the cloud rapid
adjustment and the cloud slow response under 2.5xCH$_{4SW}$ are both negative at
$-0.12 \pm 0.08$ W m$^{-2}$ and $-0.28 \pm 0.18$ W m$^{-2}$, respectively. Both are consistent
with an increase in low cloud cover (particularly the slow response at $0.31 \pm$
0.25%; Supp. Table 1). This implies that surface cooling in response to
2.5xCH$_{4SW}$ radiative effects is largely due to the cloud rapid adjustment and cloud
slow responses.
As mentioned in Section 3.1, the 2.5xCH$_{4SW}$ stratospheric temperature adjustment
under fixed climatological SSTs also significantly contributes (at $-0.04 \pm 0.01$ W
m$^{-2}$; about 1/3 the magnitude of the cloud adjustment) to the total rapid adjustment.
This negative stratospheric temperature adjustment is consistent with the relatively
large increase in stratospheric shortwave heating (Fig. 2b) and warming (Fig. 2c),
which results in enhanced outgoing longwave radiation (i.e., loss of energy and a
negative adjustment).  The tropospheric temperature adjustment (Fig. 4) is also
negative but not significant at the 90% confidence level at $-0.03 \pm 0.05$ W m$^{-2}$.
In contrast, the surface temperature adjustment at $0.02 \pm 0.01$ W m$^{-2}$ (associated
with cooling of the land surfaces and subsequent reduction in upwards longwave
radiation) acts to weakly mute the negative total rapid adjustment.  The other
2.5xCH$_{4SW}$ rapid adjustment components (e.g., tropospheric temperature, water
vapor, surface albedo) are relatively small and not significant at the 90%
confidence level.
In terms of the 2.5xCH$_{4SW}$ slow response, in addition to the dominant negative
contribution from clouds, the water vapor and surface albedo slow response also
contribute to the negative total slow response at $-0.09 \pm 0.12$ and $-0.035 \pm 0.03$
W m$^{-2}$, respectively (Fig. 4).  These are associated with tropospheric/surface
cooling, resulting in less water vapor (a greenhouse gas) and enhanced snow/ice
over land (enhanced albedo).  In contrast, the tropospheric temperature and surface
temperature slow responses are both significant and positive at $0.25 \pm 0.19$ and
$0.05 \pm 0.04$ W m$^{-2}$, respectively, and act to mute the total negative slow response
(the stratospheric temperature adjustment also weakly contributes to this muting at
$0.01 \pm 0.01$ W m$^{-2}$).
We note that the 2.5xCH$_{4SW}$ total radiative flux decomposition (sum over clouds,
water vapor, etc.) for the slow response is negative (opposite expectations since the
surface cools).  However, there is large uncertainty, i.e., it is a nonsignificant
negative value at $-0.10 \pm 0.30$ W m$^{-2}$.  This number is based on the corresponding
difference between the coupled ocean atmosphere total response and the rapid
adjustment from the fSST simulation, which have values of $-0.27 \pm 0.28$ W m$^{-2}$
and $-0.16 \pm 0.10$ W m$^{-2}$, respectively.  The former number ($-0.27 \pm 0.28$ W m$^{-2}$)
is based on the total radiative flux decomposition under 2.5xCH$_{4SW+LW}$ minus
2.5xCH$_{LW}$, which have respective values of $-0.46 \pm 0.18$ W m$^{-2}$ and $-0.19 \pm$
$0.19$ W m$^{-2}$.  So here, both values are negative, as expected (i.e., the system
responds to the positive forcing by warming and emitting more energy to space,
consistent with a stable climate system).  It is likely longer integrations (beyond 90
years) are necessary to reduce the relatively large uncertainty in some of these
values.
Decomposing the 2.5xCH$_{4SW}$ cloud rapid adjustment into shortwave and longwave
radiation components (not shown), we find the cloud rapid adjustment for
shortwave radiation is $-0.08 \pm 0.08$ W m$^{-2}$ and the cloud adjustment for longwave
radiation is $-0.05 \pm 0.03$ W m$^{-2}$.  Thus, both shortwave and longwave cloud
radiative components contribute similarly to the negative cloud rapid adjustment.
Decomposing the slow cloud response into shortwave and longwave radiation
components, we find corresponding values of $-0.33 \pm 0.17$ and $0.05 \pm 0.05$ W
m$^{-2}$, respectively.  Here, the negative cloud slow response is largely due to cloud
shortwave radiative effects (consistent with the low cloud increase of $0.31 \pm$
0.25%; Supp. Table 1), which is partially muted by cloud longwave radiative
effects.  These changes are qualitatively consistent with the 2.5xCH$_{4SW}$ CLOUD
changes discussed in Section 3.3, under the broad assumption that low clouds
primarily reflect shortwave radiation and high clouds primarily inhibit outgoing
longwave radiation. 2.5xCH$_{4SW}$ CLOUD changes under the fast response (Fig. 2e)
are augmented in the upper-troposphere (larger decreases in high-level cloud) as
compared to the total response (Fig. 3c) and in particular as compared to the slow
(Supplementary Figure 5c; Supplementary Figure 6d) response.  The weaker
decrease in upper-level clouds under the slow response is consistent with a lack of
an increase in upper-tropospheric shortwave heating rate (Supplementary Fig. 6a).
These statements are clearer under 10xCH$_{4SW}$ (Supplementary Figure 5i;
Supplementary Figure 7).
In contrast, CLOUD changes under the total response (and the slow response) are
augmented in the low to mid-troposphere (larger increases in low to mid-level
cloud) as compared to the fast response.  The larger increase in low-level cloud
under the slow response (most of which occurs over marine stratocumulus regions
off the North and South American western coasts; Supplementary Figure 5a,d,g,j)
is consistent with a low-level cloud positive feedback i.e., surface cooling
promotes more low clouds and in turn, more cooling, etc. (Clement et al., 2009;
Zelinka et al., 2020).

To summarize, we find that the shortwave absorption associated with the present-
day methane perturbation (2.5xCH$_4$) offsets 28% (7 to 55%) of the surface
warming associated with its longwave radiative effects.  Similarly, although not
significant, methane shortwave absorption associated with the present-day
perturbation mutes 19% of the positive ERF under methane longwave radiative
effects; and 66% of the precipitation increase is offset.  These responses are
associated with changes in the vertical profiles of shortwave heating (i.e., increases
for pressures < 700 hPa and decreases for pressures > 700 hPa) which impacts
atmospheric temperature, relative humidity and cloud cover.  Although some of the
2.5xCH$_{4SW}$ results lack significance at the 90% confidence level (e.g., the total
precipitation response) they are qualitatively consistent with the results based on
the larger 5xCH$_4$ and 10xCH$_4$ perturbations showed in A23 (where, for example,
the total precipitation response is significant).  The lack of more significant signals
under 2.5xCH$_{4SW}$ is due to the weaker perturbation relative to internal climate
variability.  However, the consistency of the 2.5xCH$_{4SW}$ signals relative to those
under the larger methane perturbations (5xCH$_{4SW}$ and 10xCH$_{4SW}$) supports the
robustness of the main conclusions regarding the importance of methane SW
absorption.
**3.5 Additional Analysis of the Precipitation Response**
Precipitation responses can be understood from an energetic perspective (Muller
and O'Gorman, 2011; Richardson et al., 2016; Liu et al., 2018).  Precipitation is
related to the diabatic cooling and the dry static energy flux divergence of the
atmosphere as $L_cP = Q + H$, where $L_c$ is the latent heat of condensation of water
vapor; P is precipitation; Q is the column integrated diabatic cooling of the
atmosphere excluding latent heating; and H is the column integrated dry static
energy flux divergence.  Q is estimated as LWC + SWC + SH.  LWC is the net
longwave radiative cooling of the atmosphere.  SWC is the net shortwave radiative
cooling of the atmosphere.  The "C" stands for cooling, i.e., positive SWC and
LWC represent cooling of the atmospheric column.  In CESM2, positive longwave
radiative fluxes are upwards, so LWC is calculated as the net LW radiation at the
TOA minus that at the surface. In CESM2, positive shortwave radiative fluxes are
downwards, so SWC is calculated as the net SW radiation at the surface minus the
net SW radiation at the TOA (or equivalently, the negative of the net SW radiation
at TOA minus that at the surface).  Both terms are positive for cooling (energy
loss).  SH is the downwards sensible heat flux at the surface (i.e., positive values
indicate atmospheric cooling). H is estimated as the residual between $L_cP$ and Q.
In the global mean, the circulation term (i.e., H) is zero, implying $L_cP = Q$.  As Q is
composed of LWC and SWC (and SH but it is generally small), this balance shows
that condensational heating via precipitation is largely balanced by radiative
cooling of the atmosphere.  An increase in atmospheric SW absorption (e.g., via
CH$_{4SW}$) will decrease atmospheric radiative cooling and in turn, decrease
precipitation.
Figure 5a,b shows the atmospheric energy budget decomposition for the total, fast
and slow responses under 10xCH$_{4SW}$ and 2.5xCH$_{4SW}$. Under both CH$_{4SW}$
perturbations, the decrease in global mean precipitation (i.e., the energy of
precipitation L$_c$P) is dominated by the slow response. For example, under
2.5xCH$_{4SW}$ L$_c$P decreases by $-0.09 \pm 0.09$ W m$^{-2}$ under the fast response. This
increases (in magnitude) to $-0.15 \pm 0.30$ W m$^{-2}$ under the slow response (i.e.,
total decrease is $-0.24 \pm 0.28$ W m$^{-2}$). Although these 2.5xCH$_{4SW}$ changes are not
significant at the 90% confidence level, all three L$_c$P decreases are significant
under 10xCH$_{4SW}$ at $-0.29 \pm 0.10$, $-0.83 \pm 0.27$ and $-1.12 \pm 0.25$ W m$^{-2}$,
respectively. The precipitation decrease under the slow response is largely
associated with a decrease in net longwave atmospheric radiative cooling (i.e.,
LWC) of $-0.17 \pm 0.34$ W m$^{-2}$ for 2.5xCH$_{4SW}$ and $-1.03 \pm 0.32$ W m$^{-2}$ for
10xCH$_{4SW}$ (i.e., anomalous longwave radiative warming) which is consistent with
cooling of the troposphere (e.g., Supplementary Fig. 6b and 7b). The decrease in
net longwave atmospheric radiative cooling under the slow response is weakly
muted by an increase in net shortwave radiative cooling at $0.03 \pm 0.08$ W m$^{-2}$ for
2.5xCH$_{4SW}$ and $0.30 \pm 0.09$ W m$^{-2}$ for 10xCH$_{4SW}$ (i.e., anomalous shortwave
radiative cooling), consistent with tropospheric cooling and decreases in
atmospheric water vapor (i.e., specific humidity decreases throughout the
troposphere under the slow response; Supplementary Fig. 6f and 7f). This yields
less solar absorption by water vapor, i.e., QRS decreases in the mid- and upper-
troposphere under the slow response (Supplementary Fig. 6a and 7a).
The CH$_{4SW}$ decrease in L$_c$P under the fast response is associated with opposite
changes in SWC and LWC, including dominance of the SWC term as opposed to
the LWC term. This includes a SWC decrease of $-0.18 \pm 0.03$ W m$^{-2}$ for
2.5xCH$_{4SW}$ and $-0.85 \pm 0.04$ W m$^{-2}$ for 10xCH$_{4SW}$ (i.e., less shortwave radiative
cooling), which is consistent with the enhanced solar absorption by CH$_{4SW}$ under
the fast response (e.g., Supplementary Fig. 6a and 7a). This is partially offset by
an increase in LWC, consistent with mid- to upper-tropospheric warming and
enhanced outgoing longwave radiation.
The L$_c$P decrease under the total response is associated with similar magnitude
decreases in both SWC and LWC. This is particularly true for 10xCH$_{4SW}$, where
the SWC term decreases by $-0.55 \pm 0.08$ W m$^{-2}$ and the LWC term decreases by
$-0.51 \pm 0.30$ W m$^{-2}$. Under 2.5xCH$_{4SW}$, the corresponding changes are $-0.15 \pm$
$0.07$ and $-0.08 \pm 0.33$ W m$^{-2}$, respectively. In all cases, the H term is near zero in
the global mean (i.e., energy transport in global mean should be zero). Similarly,
the SH term is generally small in all cases.
To summarize these results, the decrease in global mean precipitation under $CH_{4SW}$
is associated with both the fast and slow response, with most of the precipitation
decrease related to the slow (surface temperature mediated) response.  The
decrease in precipitation under the fast response is largely due to the enhanced
solar absorption by $CH_{4SW}$ (decrease in the SWC term above), i.e., as atmospheric
solar absorption increases, net atmospheric radiative cooling decreases, which
leads to a decrease in precipitation. In contrast, the decrease in precipitation under
the slow response is largely due to cooling of the troposphere and a decrease in net
longwave atmospheric radiative cooling (decrease in the LWC term above).
The importance of both the fast and slow response (and the dominance of the slow
response) in driving less global mean precipitation under $CH_{4SW}$ is in contrast to
other shortwave absorbers such as black carbon.  With idealized black carbon
perturbations, for example, the fast and slow global mean precipitation responses
oppose one another.  The fast response (associated with black carbon atmospheric
solar absorption) yields a global mean decrease in precipitation whereas the weaker
slow response (associated with surface warming) yields an increase in global mean
precipitation (Samset et al., 2016; Stjern et al., 2017).  The net result is a decrease
in global mean precipitation, largely due to the fast response and enhanced
atmospheric solar absorption by black carbon.
This difference in behavior between BC and $CH_{4SW}$ is because BC has a positive
TOA ERF whereas $CH_{4SW}$ has a negative TOA ERF.  The positive TOA ERF
under BC acts to warm the surface, which promotes an increase in precipitation
under the slow response.  The negative TOA ERF under $CH_{4SW}$ acts to cool the
surface (as shown here), which promotes a decrease in precipitation under the slow
response.  However, both BC and $CH_{4SW}$ have a positive atmospheric ERF (which
promotes less precipitation via fast adjustments).
Thus, the main difference between the black carbon and $CH_{4SW}$ impact on global
mean precipitation is related to the slow response.  Black carbon warms the surface
which mutes the overall decrease in global mean precipitation (from the fast
response).  In contrast, $CH_{4SW}$ cools the surface, which adds to the overall decrease
in global mean precipitation (and contributes more to the decrease than does the
fast response).
We further decompose the global mean precipitation response based on the
equation $L_c\Delta P = A + \eta\Delta TAS$ (e.g., Fläschner et al., 2016) where $L_c$ is defined above
(and equal to 29 W m$^{-2}$ (mm day$^{-1}$)$^{-1}$); $\Delta P$ is the change in the global mean
precipitation [mm day$^{-1}$]; $\Delta TAS$ is the change in global mean near-surface air
temperature [K]; A is an adjustment term (estimated from our fSST experiments)
that accounts for the change in precipitation independent of any change in surface
temperature [W m$^{-2}$], which can be further decomposed into SWC+LWC+SH,
where SWC is the net shortwave radiative cooling of the atmosphere as defined
above [W m$^{-2}$] ; LWC is the net longwave radiative cooling of the atmosphere as
defined above [W m$^{-2}$]; and SH is the downwards sensible heat flux at the surface
[W m$^{-2}$] (positive values for these three terms indicate cooling and energy loss; as
defined above). The hydrological sensitivity parameter is η [W m$^{-2}$ K$^{-1}$].
Supplementary Table 3 (and Supplementary Figure 8) shows that the hydrological
sensitivity parameter is always larger (in magnitude) under the various SW+LW
signals (e.g., 2.5xCH$_{4LW+SW}$) as compared to the LW-only signal (e.g., 2.5xCH$_{4LW}$).
The SW signal consistently (outside of 2.5xCH$_{4SW}$) yields the smallest η.
However, η has a relatively large uncertainty and it is not significantly different
between the various SW+LW signals and the corresponding LW-only signals. For
example, the hydrological sensitivity parameter is $2.47 \pm 0.24$ W m$^{-2}$ K$^{-1}$ for
10xCH$_{4LW+SW}$ and $2.39 \pm 0.16$ W m$^{-2}$ K$^{-1}$ for 10xCH$_{4LW}$. The corresponding value
for 10xCH$_{4SW}$ is $2.24 \pm 0.73$ W m$^{-2}$ K$^{-1}$. Thus, although there are systematic
differences, the hydrological sensitivity parameter is not significantly different
under the LW-only effects versus SW effects of CH$_4$.
**3.6 Comparisons with CO$_{2SW}$**
In addition to CH$_4$, other greenhouse gases (GHGs), including carbon dioxide
(CO$_2$), also absorb solar radiation.  As with most climate models, CESM2 (via
RRTMG) includes a representation of CO$_2$ SW absorption.  In particular, RRTMG
includes CO$_2$ SW absorption in four NIR/mid-IR bands: 1.3-1.6 μm, 1.9-2.15 μm,
2.5-3.1 μm and 3.8-12.2 μm. As mentioned above, RRTMG underestimates CO$_2$
SW IRF by 25-45% (Hogan and Matricardi, 2020).
Prior studies (focused on the radiative forcing) have shown the SW absorption
effects of the present-day CO$_2$ perturbation are relatively small (Myhre et al., 1998;
Etminan et al., 2016; Shine et al., 2022).  For example, from the perspective of the
SARF at the tropopause, CO$_2$ SW absorption yields a negative forcing that acts to
decrease the magnitude of the CO$_2$ LW forcing by about 5% (Myhre et al., 1998;
Etminan et al., 2016).  This is largely due to direct SW absorption in the
stratosphere dominating over relatively weak increases in tropospheric SW
absorption due to overlap with water vapor (Etminan et al., 2016).  The former acts
to decrease downward SW at the tropopause (leading to a negative contribution
that dominates the net effect), whereas the latter decreases upwards SW at the
tropopause (leading to a smaller, positive forcing). The direct SW absorption in
the stratosphere, by reducing LW cooling, also affects the temperature adjustment
(i.e., the LW flux from the stratosphere to the troposphere is increased). As shown
by Etminan et al. (2016), the overall negative contribution due to $CO_{2sw}$ is due to
the dominance of its 2.7 μm band. In contrast, for $CH_{4sw}$, the overall positive SW
forcing is due to both its 1.7 and 2.3 μm bands. This contrasting behavior between
$CO_{2SW}$ and $CH_{4SW}$ is largely driven by the amount of overlap of the SW absorption
bands with the near-IR absorption bands for water vapor (Etminan et al., 2016).

To gain a better understanding of the importance of the SW absorption effects due
to $CH_4$ relative to $CO_2$, we repeat our suite of CESM2 experiments, but based on
idealized $CO_2$ perturbations, including 2x and 4x preindustrial atmospheric $CO_2$
concentrations. This includes two sets of identical experiments (e.g., Table 1), one
that includes $CO_2$ LW+SW radiative effects (e.g., $2xCO_2^{EXP}$) and one that lacks
$CO_2$ SW radiative effects (e.g., $2xCO_{2NOSW}^{EXP}$). $CO_2$ SW absorption in the four
NIR/mid-IR bands in RRTMG is turned off in the simulations that lack $CO_2$ SW
radiative effects. These are compared to the default preindustrial control
experiment ($PIC^{EXP}$), which includes $CO_2$ (and $CH_4$) LW+SW radiative effects, as
well as to a new preindustrial control experiment with $CO_2$ SW radiative effects
turned off (i.e., LW effects only, denoted as $PIC_{NOCO2SW}^{EXP}$). As with the methane
perturbations, this suite of $CO_2$ simulations allows quantification of the $CO_2$
LW+SW, LW and SW radiative effects, denoted for example as $2xCO_{2LW+SW}$,
$2xCO_{2LW}$ and $2xCO_{2SW}$. The $2xCO_{2LW+SW}$ signal is obtained by subtracting the
default $2xCO_2$ perturbation from the default control ($2xCO_2^{EXP} - PIC^{EXP}$). The
$2xCO_{2LW}$ signal is obtained by subtracting the $2xCO_2$ perturbation without $CO_2$
SW absorption from the corresponding control simulation without $CO_2$ SW
absorption ($2xCO_{2NOSW}^{EXP} - PIC_{NOCO2SW}^{EXP}$). The $2xCO_{2SW}$ signal is obtained by
taking the double difference, i.e., $(2xCO_2^{EXP} - PIC^{EXP}) - (2xCO_{2NOSW}^{EXP} -$
$PIC_{NOCO2SW}^{EXP})$.
We note here that it is difficult to directly compare our $CH_4$ and $CO_2$ results. For
example, $2.5xCH_4$ represents an increase of ~0.0012 ppm whereas $2xCO_2$
represents an increase of ~560 ppm. Nonetheless, we provide a qualitative
comparison below.
Figure 6 shows the corresponding TOA radiative fluxes and rapid adjustments for
both $2xCO_2$ and $4xCO_2$ (Supplementary Figure 9 shows the $4xCO_{2SW}$ radiative flux
decompositions for the total, fast and slow response). As expected, these
perturbations yield a large positive TOA LW IRF at $2.59 \pm 0.05$ W m$^{-2}$ for $2xCO_2$
and $5.30 \pm 0.05$ W m$^{-2}$ for 4x$CO_2$. The corresponding TOA SW IRFs are also
positive, but they are much smaller at $0.03 \pm 0.05$ and $0.05 \pm 0.05$ W m$^{-2}$,
respectively. The total rapid adjustment for both $CO_2$ perturbations is negative
under SW radiative effects at $-0.06 \pm 0.08$ W m$^{-2}$ for 2x$CO_2$ and $-0.40 \pm 0.09$
W m$^{-2}$ for 4x$CO_2$. The larger negative total ADJ offsets the less positive IRF,
leading to a negative ERF at $-0.03 \pm 0.15$ W m$^{-2}$ for 2x$CO_{2SW}$ and $-0.35 \pm 0.15$
W m$^{-2}$ for 4x$CO_{2SW}$ (only the latter is significant at the 90% confidence level). We
reiterate that these negative values are due to isolation of $CO_2$ shortwave
absorption alone; $CO_2$'s longwave effects still dominate the total rapid adjustment
and ERF. Recall that under $CH_4$, the shortwave effects dominate the total SW+LW
rapid adjustment but not the ERF (Fig. 1).
These results are qualitatively consistent with 2.5x$CH_{4SW}$ (Fig. 1), including a
negative ADJ that offsets the positive IRF, leading to a negative ERF. The
methane SW radiative effect, however, represents a larger percentage of its LW
radiative effect. As discussed above, $CH_{4SW}$ offsets ~20% of the positive ERF
associated with $CH_{4LW}$ (although not significant under 2.5x$CH_4$). This is due to a
relatively strong negative rapid adjustment associated with $CH_{4SW}$ (e.g., $-0.16 \pm$
$0.10$ W m$^{-2}$ for 2.5x$CH_{4SW}$, which increases to $-0.77 \pm 0.11$ W m$^{-2}$ for
10x$CH_{4SW}$). This, in turn, drives the negative $CH_{4SW}$ ERF.
In contrast, 2x$CO_{2SW}$ and 4x$CO_{2SW}$ offset only 0.7% and 4%, respectively (only the
latter is significant at the 90% confidence level), of the positive ERF associated
with their LW radiative effects. The weaker $CO_{2SW}$ muting of $CO_{2LW}$ ERF is
related to a relatively weak $CO_{2SW}$ negative adjustment ($-0.06 \pm 0.08$ W m$^{-2}$ for
2x$CO_{2SW}$, but increasing to $-0.40 \pm 0.09$ W m$^{-2}$ for 4x$CO_{2SW}$), that leads to a
relatively weak negative $CO_{2SW}$ ERF. The weaker $CO_{2SW}$ muting of $CO_{2LW}$ ERF is
also related to the relatively large and positive $CO_{2LW}$ ERF. This large and positive
$CO_{2LW}$ ERF is due to a relatively large and positive ADJ under $CO_{2LW}$ (largely due
to the stratospheric temperature adjustment, as well as clouds; Fig. 6) which
reinforces the relatively large and positive $CO_{2LW}$ IRF. For example, 2x$CO_{2LW}$
yields an ADJ of $1.55 \pm 0.08$ W m$^{-2}$ and a corresponding ERF of $4.15 \pm 0.10$ W
m$^{-2}$. Thus, the weaker $CO_{2SW}$ muting of $CO_{2LW}$ ERF is related to a relatively weak
SW radiative effect, particularly compared to its very strong LW radiative effect.
We also note that the negative total rapid adjustment due to $CO_2$ SW absorption is
dominated by a negative stratospheric temperature adjustment (Fig. 6c,d). This is
also in contrast to methane, where clouds (followed by the stratospheric
temperature adjustment) drive most of the negative total rapid adjustment under
SW radiative effects (Fig. 1b). For 4x$CO_{2SW}$, the stratospheric adjustment is
$-0.46 \pm 0.01$ W m$^{-2}$ as compared to $-0.19 \pm 0.07$ W m$^{-2}$ for clouds.  This larger
negative stratospheric adjustment under 4xCO$_{2SW}$ is consistent with relatively large
shortwave heating above ~200 hPa (to be discussed below).
The ERF, IRF and ADJ under 2xCO$_2$ LW+SW radiative effects shown here
compare well with those from PDRMIP (Smith et al., 2018), although CESM2
yields a larger positive ADJ (and ERF).  For example, PDRMIP yields a multi-
model mean IRF, ERF and ADJ of ~2.5, 3.7 and 1.2 W m$^{-2}$, respectively.  The
corresponding values from our 2xCO$_2$ CESM2 simulation are $2.6 \pm 0.06$, $4.1 \pm$
$0.11$ and $1.6 \pm 0.07$ W m$^{-2}$.  The bulk of CESM2's larger ADJ is due to a larger
cloud adjustment at $0.98 \pm 0.05$ W m$^{-2}$ compared to 0.45 W m$^{-2}$ for PDRMIP.
Figure 7a shows the global mean instantaneous shortwave heating rate profile for
2xCO$_{2SW}$ and 4xCO$_{2SW}$.  Both profiles show a decrease in QRS throughout the
troposphere with two minima, one near 800 hPa in the lower-troposphere and
another near 250 hPa in the upper troposphere.  Above 200 hPa, QRS increases
rapidly through the stratosphere, reaching ~0.15 K d$^{-1}$ at 3.6 hPa under 4xCO$_{2SW}$.
The vertical structure of QRS under CO$_{2SW}$ shows similarities to that under CH$_{4SW}$
(Fig. 2a), but CO$_{2SW}$ exhibits QRS decreases throughout the entire troposphere as
well as relatively large QRS increases in the stratosphere.  In other words, the
transition level from decreasing to increasing QRS occurs higher aloft under
CO$_{2SW}$, with larger QRS increases in the stratosphere.
The corresponding fSST 'fast' responses are included in Figure 7b-f.  The QRS
profile (Fig. 7b) is very similar to the corresponding instantaneous profile (Fig. 7a).
The relatively large CO$_{2SW}$ stratospheric solar heating helps to explain the
correspondingly large negative stratospheric temperature adjustment (Fig. 6c,d).
That is, the large increase in stratospheric solar absorption leads to corresponding
warming and subsequently, enhanced outgoing longwave radiation which acts to
cool the climate system.  The decrease in tropospheric QRS is associated with
weak cooling (Fig. 7c), and increases in both relative humidity (Fig. 7d) and clouds
(Fig. 7e), with stronger responses under 4xCO$_{2SW}$ as compared to 2xCO$_{2SW}$.  The
opposite responses occur in the stratosphere.  These results again share similarities
to those based on CH$_{4SW}$ (Fig. 2), but CO$_{2SW}$ exhibits more uniform changes
throughout the troposphere (i.e., the transition level occurs higher aloft), as well as
relatively large stratospheric changes.
Due to the relatively weak and non-significant 2xCO$_{2SW}$ radiative fluxes (and
limited computational resources), we only perform the coupled ocean-atmosphere
simulations for 4xCO$_2$.  Figure 8a-c shows the global mean total, fast and slow
response vertical profiles under 4xCO$_{2SW}$ for QRS, temperature and cloud cover.
Significant cooling (Fig. 8b) occurs under the total (and slow) response throughout
the troposphere, with maximum cooling of ~0.5 K near 200 hPa under the total
response.  Above this level, cooling gradually weakens and transitions into
warming aloft, peaking at ~1 K near 50 hPa. The corresponding vertical CLOUD
total response profile (Fig. 8c) shows increasing cloud cover throughout the
troposphere, with decreases aloft (near 100 hPa), generally similar to the fast
response but with larger tropospheric CLOUD increases and weaker CLOUD
decreases aloft.  The global maps of the TAS and P total climate response under
$4xCO_{2SW}$ are included in Figure 8d,e.  $4xCO_{2SW}$ drives a significant decrease in
TAS and P at $-0.38 \pm 0.12$ K and $-0.031 \pm 0.01$ mm d$^{-1}$ (-1.05%).
Supplementary Table 2 (and Supplementary Figure 3d) show the individual
components of the TOA energy decomposition equation, including the estimated
climate feedback parameter, for the $4xCO_2$ simulations.  As with the methane
signals, the climate feedback parameter is larger (in magnitude) under $4xCO_{2LW+SW}$
as compared to $4xCO_{2LW}$, but not significantly so. For example, $\alpha$ is $-1.18 \pm 0.06$
W m$^{-2}$ K$^{-1}$ for $4xCO_{2LW+SW}$ and $-1.11 \pm 0.06$ W m$^{-2}$ K$^{-1}$ for $4xCO_{2LW}$.  The
corresponding $\alpha$ value for $4xCO_{2SW}$ is $-0.31 \pm 0.93$ W m$^{-2}$ K$^{-1}$.
Under $4xCO_{2SW}$, the TAS and P responses are quite small as compared to the
corresponding LW radiative effects at $5.84 \pm 0.08$ K and $0.27 \pm 0.01$ mm d$^{-1}$
(9.1%), respectively.  For example, if $CH_{4LW}$ yielded the same 5.84 K of warming,
this would correspond to surface cooling associated with $CH_{4SW}$ of ~1.75K
(assuming 30% offset, which may not apply here). In terms of TAS, $4xCO_{2SW}$
mutes 6.5% of the warming due to LW radiative effects.  For P, $4xCO_{2SW}$ mutes
11.5% of the increase in precipitation due to LW radiative effects.  Thus, the
muting effects of $CO_{2SW}$ are much weaker than those associated with $CH_{4SW}$, where
~30% of the warming and ~60% of the wetting due to $CH_4$ LW radiative effects
are offset.
We also perform the atmospheric energy balance calculation (Section 3.5) on the
suite of $4xCO_{2SW}$ simulations (Fig. 5c).  Overall, the conclusions discussed in
Section 3.5 under $2.5xCH_{4SW}$ and $10xCH_{4SW}$ also apply under $4xCO_{2SW}$.  The
decrease in the global mean energy of precipitation under $4xCO_{2SW}$ ($-0.91 \pm 0.30$
W m$^{-2}$ under the total response) is associated with both the fast (a non-significant
decrease of $-0.08 \pm 0.11$ W m$^{-2}$) and slow response ($-0.83 \pm 0.32$ W m$^{-2}$).  Here,
nearly all of the precipitation decrease (91% as opposed to 63% for $2.5xCH_{4SW}$ and
74% for $10xCH_{4SW}$) is related to the slow (surface temperature mediated) response.
In other words, only 9% of the precipitation decrease under $4xCO_{2SW}$ is due to the
fast response, which is much lower than that under $CH_{4SW}$ (26-37%).  The weaker
contribution to the decrease in total precipitation by the $4xCO_{2SW}$ fast response is
consistent with similar (but opposite signed) changes in the SWC and LWC terms
at $-0.41 \pm 0.04$ W m$^{-2}$ and $0.35 \pm 0.12$ W m$^{-2}$, respectively, which neutralize one
another. This cancellation is consistent with the $4xCO_{2SW}$ solar heating profile
(e.g., Fig. 7b) where nearly all of the heating occurs in the stratosphere. Thus, the
added solar heating—although decreasing the SWC term—primarily warms the
stratosphere where the energy is efficiently radiated back to space (i.e., the SWC
decrease is primarily balanced by an increase in the LWC term). This is in contrast
to the QRS profiles under $CH_{4SW}$ (e.g., Fig. 2b) which show significant solar
absorption throughout the mid- and upper troposphere (pressures $< 700$ hPa).
Thus, we suggest the relatively weak decrease in precipitation under the $4xCO_{2SW}$
fast response (relative to the $CH_{4SW}$ perturbations) is related to differences in the
vertical QRS profile, with $CO_{2SW}$ solar absorption primarily occurring in the
stratosphere.
Supplementary Table 3 (and Supplementary Figure 8d) show the individual
components of the alternate precipitation energy decomposition equation,
including the estimated hydrological sensitivity parameter, for the $4xCO_2$
simulations. For example, η is $2.47 \pm 0.04$ W m$^{-2}$ K$^{-1}$ for $4xCO_{2LW+SW}$ and $2.46 \pm$
$0.04$ W m$^{-2}$ K$^{-1}$ for $4xCO_{2LW}$. The corresponding η value for $4xCO_{2SW}$ is smaller
(but not significantly so, as with methane) at $2.31 \pm 0.89$ W m$^{-2}$ K$^{-1}$. Thus, similar
to the methane simulations, although there are systematic differences, we do not
find significant differences between the hydrological sensitivity parameter under
the LW-only effects versus the SW effects of $CO_2$.
**3.7 Climate Feedbacks**
As discussed above, the climate feedback parameter (as estimated via a regression
approach; Supp. Table 2) is always larger (in magnitude) under the various
SW+LW signals (e.g., $2.5xCH_{4LW+SW}$) as compared to the LW-only signal (e.g.,
$2.5xCH_{4LW}$). Although these differences are not significant, they suggest the
climate system does not have to warm as much to offset the same TOA energy
imbalance when SW effects are included. We perform an alternate procedure to
calculate the total climate feedback and its components by normalizing the slow
response's radiative flux decomposition (based on the radiative kernel method) by
the corresponding change in global mean near-surface air temperature. Figure 9
shows the corresponding feedback decomposition. We first point out that the total
climate feedback as calculated here ($\alpha_k$) is similar (i.e., error bars overlap except
for $4xCO_2$) to that previously estimated using the regression approach ($\alpha$) (Supp.
Table 2). Thus, $\alpha_k$ is also always larger (in magnitude) under the various
SW+LW signals as compared to the corresponding LW-only signals, with
consistently smaller (negative) magnitudes under the SW-only signals (outside of
2.5xCH$_{4SW}$).  Although $\alpha_k$ has smaller uncertainty (as compared to $\alpha$), these
differences continue to lack significance (i.e., blue bar's errors overlap in Fig. 9).
It is also clear, however, that the individual feedbacks (e.g., tropospheric
temperature feedback) are all very similar across CH$_4$ and CO$_2$ LW+SW, LW and
SW radiative effects—except the cloud feedback, where significant differences
exist (for the larger perturbations).  For example, the cloud feedback is $0.05 \pm$
$0.20$ W m$^{-2}$ K$^{-1}$ for 10xCH$_{4LW+SW}$; $0.36 \pm 0.09$ W m$^{-2}$ K$^{-1}$ for 10xCH$_{4LW}$; and $1.0 \pm$
$0.53$ W m$^{-2}$ K$^{-1}$ for 10xCH$_{4SW}$ (i.e., the cloud feedback is significantly different
between SW versus LW radiative effects; Fig. 9a).  Thus, the larger (positive)
cloud feedback under SW radiative effects acts to weaken the total (negative)
feedback, which helps to explain the previously mentioned systematically smaller
(in magnitude) values for $\alpha$ (and $\alpha_k$) under SW effects.  Furthermore, the
systematically larger (negative) values for $\alpha$ and $\alpha_k$ under SW+LW effects is due
to a relatively weak cloud feedback (e.g., $0.05 \pm 0.20$ W m$^{-2}$ K$^{-1}$ for
10xCH$_{4LW+SW)}$.  We also clarify here that this weak cloud feedback under SW+LW
effects is due to the fact LW effects are associated with surface warming and
decreased low cloud cover under the slow response (Supp. Table 1), which in turn
drives more warming (i.e., a positive cloud feedback). This is weakened by SW
effects, which are associated with surface cooling and increased low cloud cover
under the slow response (Supp. Table 1), which in turn drives more cooling (i.e., a
positive feedback that opposes that under LW effects).  Even though the surface
cooling under SW effects is relatively small compared to the warming under LW
effects, the cloud feedback under SW effects is larger than that under LW effects,
effectively leading to a smaller cloud feedback under SW+LW effects (and not
significant under all of the CH$_4$ perturbations).  The net effect is that the planet
does not need warm up as much under SW+LW effects to restore energy balance,
due to the SW effects on clouds under the slow response (and in particular,
increased low clouds; Supp. Table 1).  Analogously, these results imply relatively
large cooling per unit forcing under methane shortwave radiative effects, which in
turns leads to relatively less warming per unit forcing under methane shortwave
and longwave radiative effects.
The importance of low clouds is further supported by an analogous feedback
decomposition that separates TOA radiative fluxes into shortwave (Supplementary
Figure 10) versus longwave fluxes (Supplementary Figure 11).  Here, the total
feedback (and individual feedbacks, including clouds) for TOA longwave fluxes is
very similar across SW+LW, LW and SW effects for each perturbation.  In
contrast, the total feedback for TOA shortwave fluxes is more positive under CH$_4$
and $CO_2$ SW effects (significantly so for the larger perturbations), and this is
driven by the cloud feedback (Supp. Fig. 10). For example, the total TOA
shortwave flux feedback is $0.45 \pm 0.21$ W m$^{-2}$ K$^{-1}$ for $10xCH_{4LW+SW}$; $0.86 \pm 0.10$
W m$^{-2}$ K$^{-1}$ for $10xCH_{4LW}$; and $1.69 \pm 0.55$ W m$^{-2}$ K$^{-1}$ for $10xCH_{4SW}$. These
differences are largely due to the corresponding cloud feedback at $-0.14 \pm 0.20$
W m$^{-2}$ K$^{-1}$ for $10xCH_{4LW+SW}$; $0.26 \pm 0.09$ W m$^{-2}$ K$^{-1}$ for $10xCH_{4LW}$; and $1.08 \pm$
$0.55$ W m$^{-2}$ K$^{-1}$ for $10xCH_{4SW}$.
Finally, we note that this cloud feedback (and its impact on the total feedback)
under SW effects is more important under $CH_4$ as opposed to $CO_2$ (Fig. 9d). For
example, although the cloud feedback is $0.85 \pm 0.32$ W m$^{-2}$ K$^{-1}$ for $4xCO_{2SW}$
(significantly different than that for $4xCO_{2LW}$), very similar values occur for
$4xCO_{2LW+SW}$ ($0.51 \pm 0.02$ W m$^{-2}$ K$^{-1}$) and $4xCO_{2LW}$ ($0.54 \pm 0.03$ W m$^{-2}$ K$^{-1}$). This
is consistent with the weaker absorption of solar radiation by $CO_2$ (relative to
$CH_4$).
**4 Discussion and Conclusions**
We have expanded upon the work of A23, by explicitly simulating the radiative
and climate responses of the present-day (2.5x preindustrial) perturbation of
methane, decomposed into LW+SW, LW and SW radiative effects. Our results
here based on $2.5xCH_4$ are consistent with the conclusions from A23, and re-
emphasize the importance of methane SW absorption—not only under relatively
large perturbations, but also under realistic, present-day perturbations (albeit with
larger uncertainty).
$2.5xCH_{4SW}$ cools the surface by $-0.10 \pm 0.07$ K whereas $2.5xCH_{4LW}$ warms the
surface by $0.35 \pm 0.05$ K. That is, $2.5xCH_{4SW}$ acts to mute 28% (7-55%) of the
warming due to the corresponding methane longwave radiative effects. Although
similar conclusions apply for precipitation, where 66% of the precipitation increase
associated with methane longwave radiative effects under the present-day methane
perturbation is offset by shortwave absorption, this muting effect is not significant
at the 90% confidence level (i.e., the global mean precipitation response under
$2.5xCH_{4SW}$ is not significant at $-0.008 \pm 0.009$ mm d$^{-1}$). Nonetheless, similar to
the larger methane perturbations emphasized in A23, SW absorption due to the
present-day $CH_4$ perturbation offsets ~30% of the warming and ~60% of the
precipitation increase associated with the present-day $CH_4$ LW radiative effects.
Muting of warming and wetting is consistent with a negative $CH_{4SW}$ ERF due to a
negative rapid adjustment dominated by clouds. This in turn weakens the positive
ERF associated with $CH_{4LW}$. Under the present-day methane perturbation, ~20%
of the ERF associated with methane longwave radiative effects is muted by
shortwave absorption, which is again similar to (but not significant here) the larger
$CH_4$ perturbations in A23.
An atmospheric energy budget analysis (Fig. 5) shows that the decrease in global
mean precipitation under $CH_{4SW}$ is associated with both the fast and slow response,
with most of the precipitation decrease related to the slow (surface temperature
mediated) response. The decrease in precipitation under the fast response is
largely due to the enhanced solar absorption by $CH_{4SW}$, whereas the decrease in
precipitation under the slow response is largely due to cooling of the
surface/troposphere and a decrease in net longwave atmospheric radiative cooling.
The importance of both the fast and slow response (and the dominance of the slow
response) in driving less global mean precipitation under $CH_{4SW}$ is in contrast to
other shortwave absorbers such as black carbon (where the fast and slow
precipitation response oppose one another).
This difference in behavior (i.e., slow precipitation response) between $CH_{4sw}$ and
BC comes from the different signs of the global temperature response which is
driven by the ERF. $CH_{4SW}$ yields a negative ERF (Fig. 1a) and surface cooling
(Fig. 3f), whereas BC yields a positive ERF and surface warming (e.g., Stjern et
al., 2017). The former surface cooling promotes a precipitation decrease whereas
the latter surface warming promotes a precipitation increase. We note that the
different signed ERFs between $CH_{4sw}$ and BC may (in part) be related to
differences in their vertical QRS profile (e.g., Allen et al., 2019). The negative
QRS in the lower troposphere promotes a negative low cloud adjustment for $CH_{4sw}$
which contributes to the negative ERF. Whereas for BC (where the QRS profile is
more vertically uniform with increases throughout the atmosphere e.g.,
Supplementary Figure 4 from Stjern et al., 2017), the positive QRS in the lower
troposphere leads to less low cloud adjustment so the ERF is overall more positive.
BC is also a stronger SW absorber than is methane (i.e., in terms of its IRF), which
also contributes to the larger positive ERF of BC.
As many climate models lack methane SW absorption, our results imply that such
models may overestimate the warming and wetting due to the increase in
atmospheric methane concentrations over the historical time period. Similarly,
such models may also have deficient simulation of the corresponding methane
climate impacts under future climate projections.
We further show the importance of $CH_{4sw}$ by comparison to $CO_{2sw}$. $CO_2$ SW
absorption yields qualitatively similar results to $CH_4$ SW absorption, including a

negative ADJ that offsets the positive IRF, leading to a negative ERF (Fig. 6; we reiterate that these negative ADJ and ERF values are due to isolation of shortwave effects alone). In contrast to $CH_{4SW}$ (where the cloud adjustment dominates), the negative ADJ under $CO_{2SW}$ is largely due to the stratospheric temperature adjustment, which is consistent with larger SW absorption in the stratosphere under $CO_{2SW}$ (Fig. 7a,b). The reduced importance of the cloud adjustment under $CO_{2SW}$ as compared to $CH_{4SW}$ is related to differences in their vertical QRS profiles. Under $CO_{2SW}$, the vertical QRS profile exhibits more vertically uniform tropospheric changes (Fig. 7a-b), with the transition level from decreasing to increasing QRS occurring higher aloft (as compared to $CH_{4SW}$; Fig. 2a,b). These QRS differences also impact the fast precipitation response (a decrease), which is less important under $CO_{2SW}$ as compared to $CH_{4SW}$ (Fig. 5). Under $CO_{2SW}$, LWC and SWC are nearly equal and opposite in sign (leading to cancellation and small precipitation changes), whereas decreases in SWC dominate over increases in LWC under $CH_{4SW}$, which promotes a precipitation decrease. As most of the atmospheric solar heating under $CO_{2SW}$ occurs in the stratosphere, this primarily warms the stratosphere where the energy is efficiently radiated back to space (i.e. the SWC decrease is primarily balanced by an LWC increase). Finally, consistent with the relatively small (negative) $CO_{2sw}$ ERF relative to the much larger positive $CO_{2LW}$ ERF, $4xCO_{2SW}$ muting of the $4xCO_{2LW}$ climate responses (e.g., temperature, precipitation) are also relatively small and about five times smaller as compared to the $2.5xCH_{4SW}$ muting effects.

Additional analysis of the climate feedback parameter α, climate sensitivity λ, and the hydrological sensitivity parameter η indicate consistent but non-significant differences between the LW and SW effects for both $CH_4$ and $CO_2$ (e.g., Supplementary Tables 2-3; Supplementary Figures 3 & 8). For example, SW effects (outside of $2.5xCH_{4SW}$) consistently yield smaller (negative) α values (and in turn larger positive λ); and smaller (positive) η. Again, however, these differences are not significant. An alternate procedure (based on radiative kernels applied to the slow response) to derive the climate feedback parameter and its components yields similar results, and also shows the importance of $CH_{4SW}$ (and to a lesser extent $CO_{2SW}$) to the cloud feedback (Fig. 9; Supp. Fig. 10-11). In particular, SW effects lead to a stronger (positive) cloud feedback (largely due to low clouds) which effectively mutes the cloud feedback under LW effects. The leads to a more negative total climate feedback when SW effects are included, implying the climate system does not need to warm up as much to restore energy balance. Analogously, these results imply relatively large cooling per unit forcing under methane shortwave radiative effects, which in turns leads to relatively less

warming per unit forcing under methane shortwave and longwave radiative effects.

Such potential differences in these parameters under SW versus LW effects
deserves additional analysis.  For example, it would be interesting to repeat some
of our simulations (particularly the larger perturbations) over a longer integration
time-period (e.g., 150+ years), which would help increase the signal to noise ratio.
Moreover, one could reassess the above climate parameters using alternative
procedures, e.g., a "Gregory"-style regression methodology (Gregory et al., 2004).
Similar simulations with multiple models would also be useful.

As our conclusions continue to be derived from one climate model, we encourage
additional multi-model studies to evaluate the robustness of these results.  Ideally,
this includes simulations that include interactive chemistry (e.g., methane can
enhance tropospheric ozone production), as our CESM2/CAM6 simulations do not.
We also reiterate that there are known deficiencies in the shortwave radiative
transfer code used in most climate model calculations, including CESM2.  As
mentioned above, CESM2's radiative transfer model (RRTMG) underestimates
$CH_4$ (and $CO_2$) SW IRF by 25-45% (Hogan and Matricardi, 2020).  This is in
addition to the various subtleties in the quantification of methane shortwave
forcing identified by Byrom and Shine (2022). These subtleties include the need
for careful representation of the spectral variation of surface albedo and the vertical
profile of methane, and the role of shortwave absorption at longer wavelengths,
specifically methane's 7.6 μm band that is not included in some climate model
radiation codes, including RRMTG.  Thus, additional efforts are needed to
improve climate model representation of $CH_{4SW}$.

In the context of the most recent IPCC ERF estimates, methane SW absorption is
included and is based on Smith et al. (2018).  The corresponding 1750-2019 (729.2
to 1866.3 ppb, or 2.6x increase) methane ERF is $0.54 \pm 0.11$ W m$^{-2}$, which includes
a correction associated with methane SW absorption of -0.08 W m$^{-2}$ (Forster et al.,
2021).  Our ERF estimate for 2.5x$CH_4$ is within this uncertainty range at $0.43 \pm$
$0.08$ W m$^{-2}$.  Furthermore, we estimate the $CH_{4SW}$ correction (i.e., the $CH_{4SW}$ ERF)
at $-0.10 \pm 0.13$ W m$^{-2}$, which compares very well to the IPCC estimate of -0.08
W m$^{-2}$.  We note that the IPCC estimate is based on four models, one of which is
CESM1 (the predecessor to the model used here). The most recent IPCC global
warming potentials (GWP) for methane (e.g., $82.5 \pm 25.8$ for fossil-$CH_4$ and a 20-
year time horizon) also include methane SW absorption.  Given the caveats
discussed above (e.g., underestimation of $CH_4$ SW IRF by 25-45%), however,
these estimates of the $CH_{4SW}$ adjustment and the corresponding climate effects may
be underestimated.
We also iterate that these are concentration ("abundance") based ERF estimates.
The methane concentration used to derive such a concentration-based ERF is based
on the observed change, which is influenced not only by the change in methane
emissions, but also changes in emissions of other compounds that affect methane
lifetime and concentrations (Stevenson et al., 2020). For example, changes in non-
methane ozone precursors including nitrogen oxides and volatile organic
compounds in general reduce methane concentrations. This means that the
methane perturbation applied here is smaller than that which would arise if
methane is emissions-driven. In the latter case, the derived methane concentration
change would be higher than that observed, would take account of the impact of
methane on its own lifetime, and would be attributable to the change in methane
emissions alone. For example, Shindell et al. (2005) shows that the instantaneous
tropopause direct radiative forcing (1998 relative to preindustrial) of methane
alone increases from 0.48 to 0.59 W m$^{-2}$, in switching from a concentration-based
to an emissions-based perspective. Accounting for the impacts of methane on
ozone production and stratospheric water vapor further increases methane's
radiative forcing to ~0.9 W m$^{-2}$ (Shindell et al., 2005). A more recent estimate of
the emissions-based methane ERF (including indirect effects) is 1.19$\pm$0.38 W m$^{-2}$
(Szopa et al., 2021). This is due to indirect positive ERFs from methane enhancing
its own lifetime, enhancing stratospheric water vapor, causing ozone production,
and influencing aerosols and the lifetimes of hydrochlorofluorocarbons (HCFCs)
and hydrofluorocarbons (HFCs) (Myhre et al., 2013; O'Connor et al., 2022). We
reiterate that our simulations do not include these methane indirect effects. Such
effects not only impact the ERF, but also the temperature response in the
stratosphere and upper troposphere (Winterstein et al., 2019), which in turn may
impact the cloud response.
In conclusion, the present-day methane perturbation is associated with $CH_{4SW}$
muting of 28% (7-55%) of the $CH_{4LW}$ surface warming. This is consistent with the
negative ERF and perhaps also a relatively strong low cloud feedback under
$CH_{4SW}$. Despite our main conclusions, we emphasize that methane remains a potent
GHG. Continued efforts to reduce $CH_4$ emissions are vital for staying below 1.5°C
of global warming.
**Code Availability**

CESM2 can be downloaded from NCAR at
https://www.cesm.ucar.edu/models/cesm2/download.  The Python-based radiative
kernel toolkit and the GFDL radiative kernel can be downloaded from
https://climate.rsmas.miami.edu/data/radiative-kernels/.

**Data Availability**

A core set of model data from the 2.5x preindustrial methane CESM2 simulations
is available here: https://doi.org/10.5281/zenodo.10357888.

**Author Contributions**

R.J.A performed CESM2/CAM6 simulations and analyzed the results.  All authors,
including X.Z., C.A.R., C.J.S., R.J.K and B.H.S discussed the results and
contributed to the writing.

**Competing Interests**

The authors declare no competing interests.

**Acknowledgements**

R. J. Allen is supported by NSF grant AGS-2153486.  We would like to
acknowledge high-performance computing support from Cheyenne
(doi:10.5065/D6RX99HX) provided by NCAR's Computational and Information
Systems Laboratory, sponsored by the National Science Foundation.  We also
acknowledge helpful comments and discussions with Keith Shine.  We thank one
anonymous reviewer and William Collins for reviewing this manuscript.

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

**Tables**

**Table 1. Description of CESM2/CAM6 methane and carbon dioxide experiments**. Both fixed climatological sea surface temperature and coupled ocean atmosphere simulations are performed for each experiment. 2.5x preindustrial atmospheric methane concentrations represent the present-day methane perturbation which corresponds to a ~750 to ~1900 ppb increase (i.e., ~150%). Analogous experiments are conducted for $2xCO_2$ and $4xCO_2$.

| Experiment | Description |
|---|---|
| $2.5xCH_4^{EXP}$ | $2.5xCH_4$ with $CH_4$ LW+SW radiative effects |
| $2.5xCH_{4NOSW}^{EXP}$ | $2.5xCH_4$ with $CH_4$ SW radiative effects turned off |
| $PIC^{EXP}$ | Preindustrial $CH_4$ with $CH_4$ LW+SW radiative effects |
| $PIC_{NOCH4SW}^{EXP}$ | Preindustrial $CH_4$ with $CH_4$ SW radiative effects turned off |
| **Signal** | **Description** |
| $2.5xCH_{4LW+SW} = 2.5xCH_4^{EXP} - PIC^{EXP}$ | Response to $CH_4$ LW+SW radiative effects |
| $2.5xCH_{4LW} = 2.5xCH_{4NOSW}^{EXP} - PIC_{NOCH4SW}^{EXP}$ | Response to $CH_4$ LW radiative effects |
| $2.5xCH_{4SW} = (2.5xCH_4^{EXP} - PIC^{EXP}) - (2.5xCH_{4NOSW}^{EXP} - PIC_{NOCH4SW}^{EXP})$ | Response to $CH_4$ SW radiative effects |

**Figures**

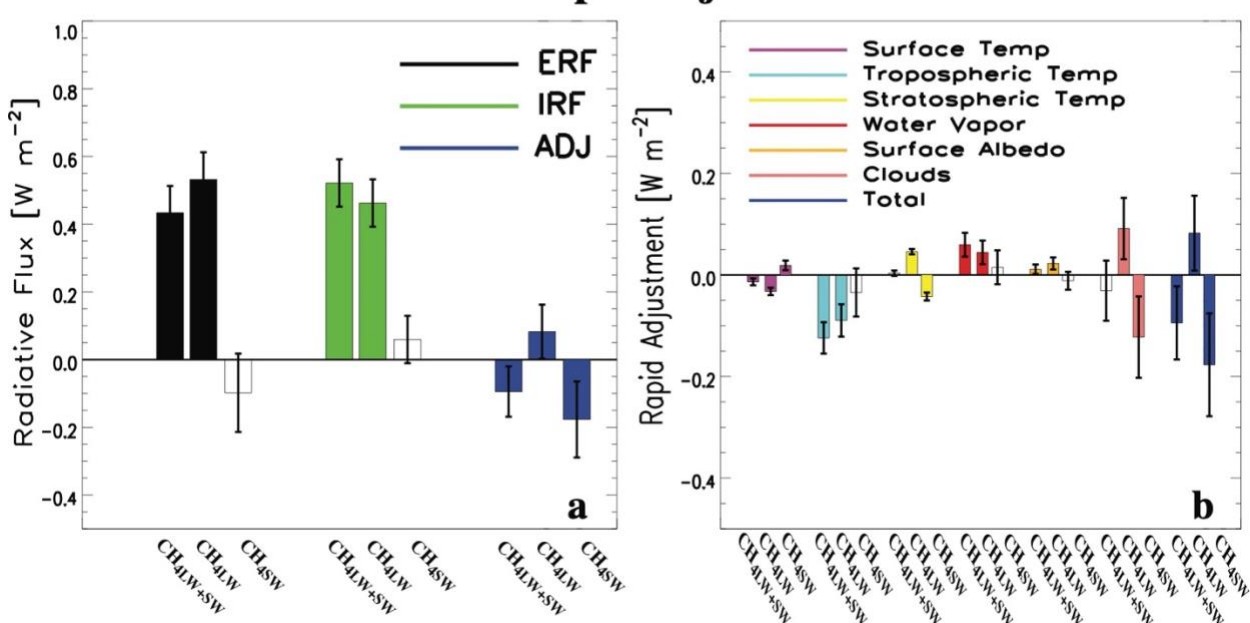

**Figure 1. Top-of-the-atmosphere radiative flux components and rapid adjustments for 2.5xCH4.** Global annual mean top-of-the-atmosphere (TOA) (a) effective radiative forcing (ERF; black), instantaneous radiative forcing (IRF; green) and rapid adjustment (ADJ; blue); and (b) decomposition of the rapid adjustment into its components including surface temperature (purple), tropospheric temperature (cyan), stratospheric temperature (yellow), water vapor (red), surface albedo (orange), cloud (pink) and total rapid adjustment (blue) for 2.5xCH4. Responses are decomposed into methane longwave and shortwave radiative effects ($CH_{4LW+SW}$), methane longwave radiative effects ($CH_{4LW}$) and methane shortwave radiative effects ($CH_{4SW}$). ERF and rapid adjustments are based on 30-year fixed climatological sea surface temperature simulations. Uncertainty is quantified using the 90% confidence interval; unfilled bars denote responses that are not significant at the 90% confidence level. Units are W m$^{-2}$.

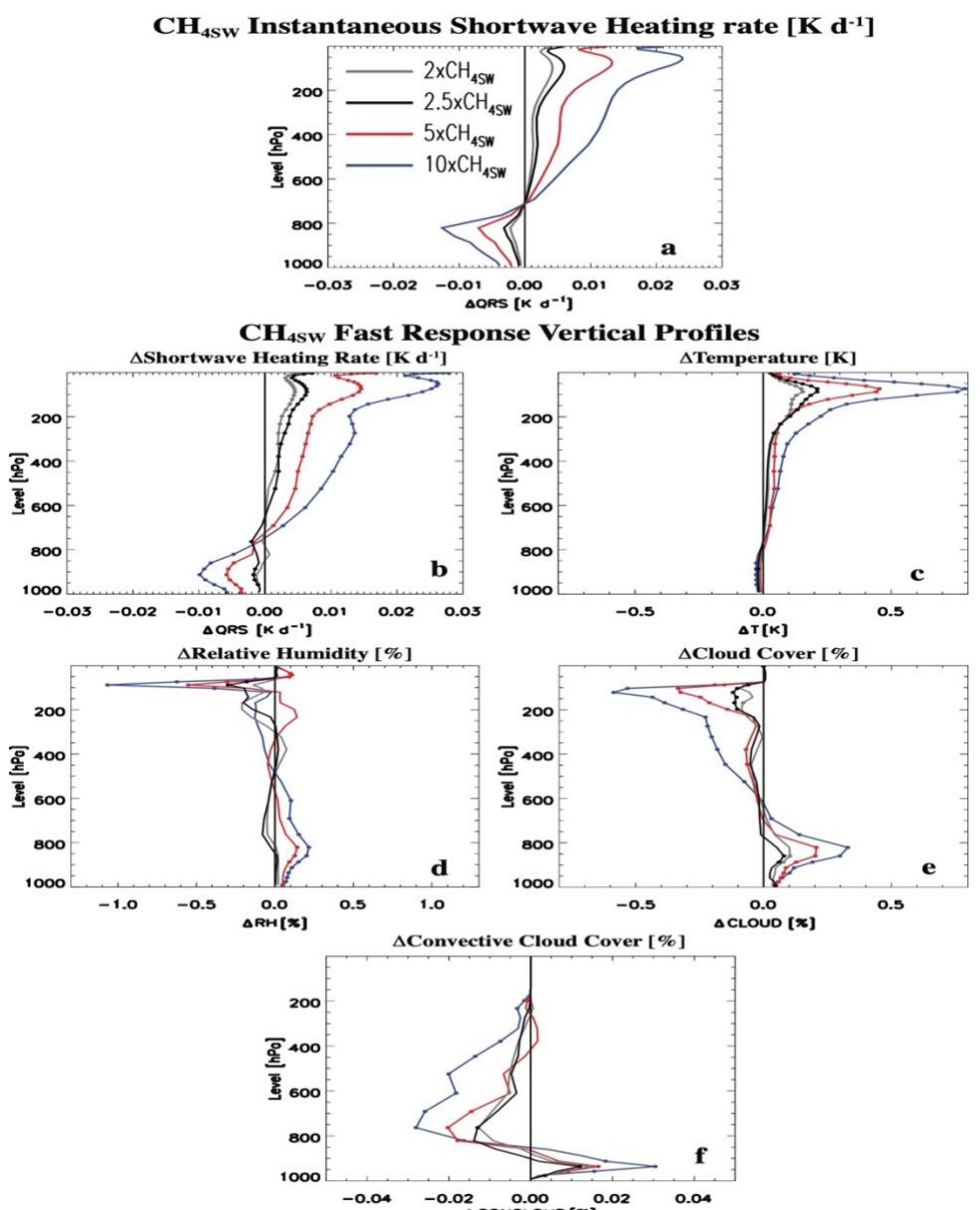

1384
**Figure 2. Global mean annual mean vertical response profiles for four CH$_{4SW}$
perturbations**. Instantaneous (a) shortwave heating rate (QRS; units are K d$^{-1}$);
and (b-f) fast responses of (b) QRS (units are K d$^{-1}$); (c) air temperature (T; units
are K); (d) relative humidity (RH; units are %); (e) cloud cover (CLOUD; units are
%) and (f) convective cloud cover (CONCLOUD; units are %) for 2xCH$_{4SW}$
(gray); 2.5xCH$_{4SW}$ (black); 5xCH$_{4SW}$ (red); and 10xCH$_{4SW}$ (blue). The 2xCH$_4$,
5xCH$_4$ and 10xCH$_4$ simulations are from A23.  A significant response at the 90%
confidence level, based on a standard t-test, is denoted by solid dots in (b-f).
Climatologically fixed SST simulations are used to estimate the fast responses.
Instantaneous QRS profiles come from the Parallel Offline Radiative Transfer
Model (PORT).

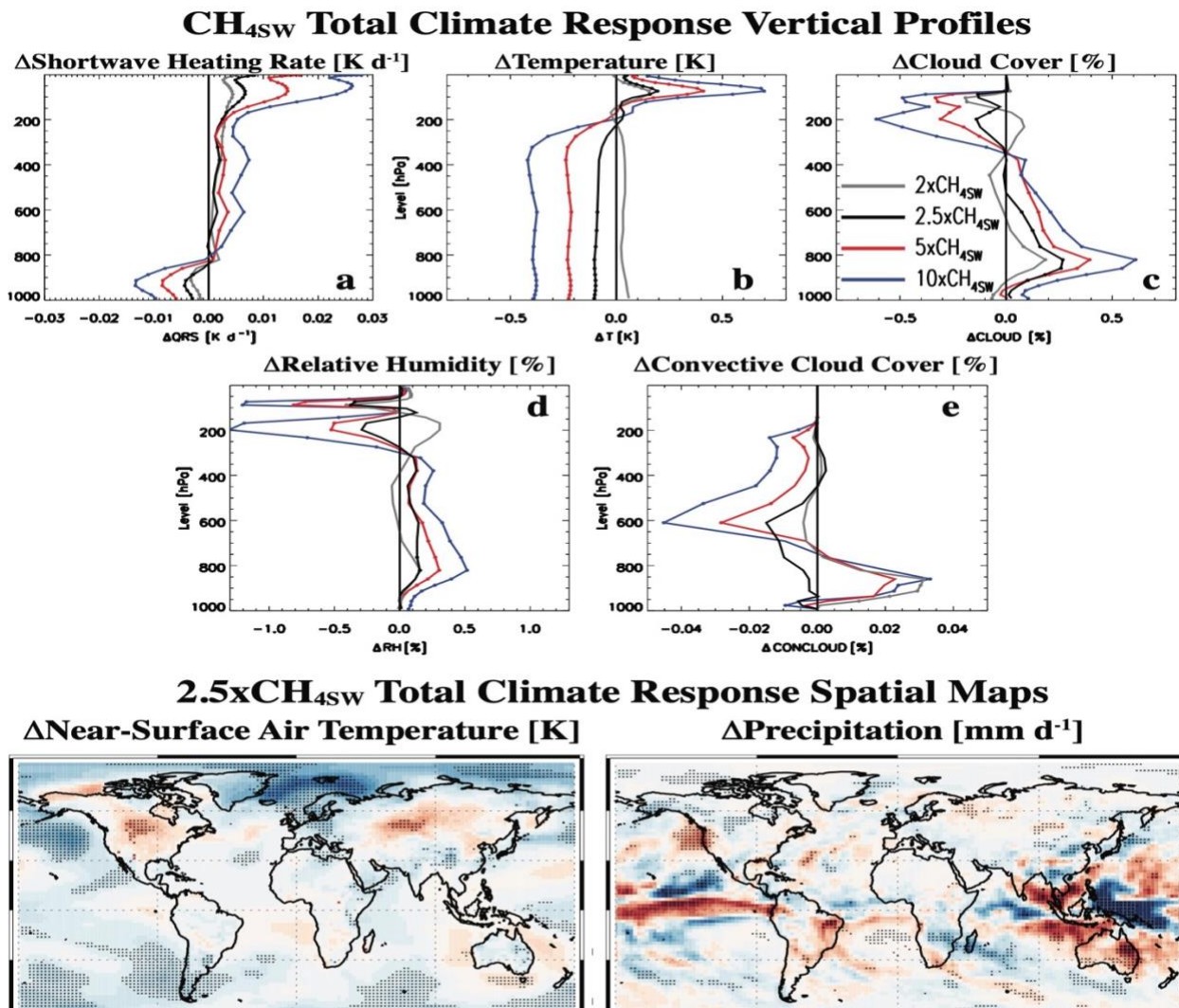

**Figure 3. Total climate responses to CH₄sw**. Annual mean global mean vertical response profiles of (a) shortwave heating rate (QRS; units are K d⁻¹); (b) air temperature (T; units are K); (c) cloud cover (CLOUD; units are %); (d) relative humidity (RH; units are %); and (e) convective cloud cover (CONCLOUD; units are %) for 2xCH₄sw (gray); 2.5xCH₄sw (black); 5xCH₄sw (red); and 10xCH₄sw (blue). The 2xCH₄sw, 5xCH₄sw and 10xCH₄sw simulations are from A23. Also included are global maps of the annual mean (f) near-surface air temperature [K] and (g) precipitation [mm d⁻¹] response for 2.5xCH₄sw. A significant response at the 90% confidence level, based on a standard t-test, is denoted by solid dots. Climate responses are estimated from coupled ocean-atmosphere CESM2 simulations.

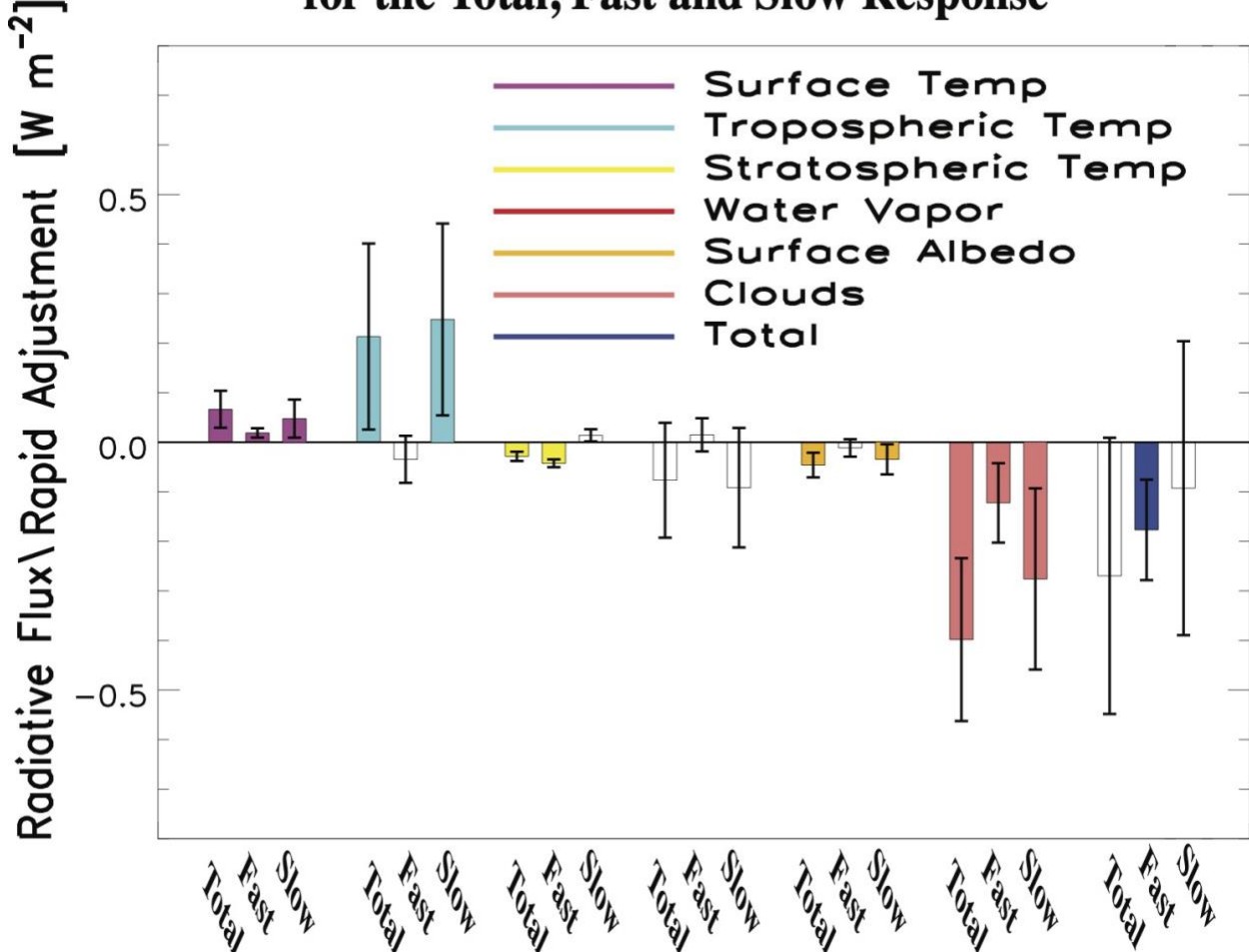

**Figure 4. 2.5xCH4sw top-of-the-atmosphere radiative flux decomposition for the total response, fast response (rapid adjustment) and slow response.** Global annual mean top-of-the-atmosphere (TOA) surface temperature (purple), tropospheric temperature (cyan), stratospheric temperature (yellow), water vapor (red), surface albedo (orange), cloud (pink) and total (blue) radiative flux decomposition for 2.5xCH4sw. The total response (from the coupled ocean atmosphere simulations) is represented by the first bar in each like-colored set of three bars; the rapid adjustment (fast response from fixed climatological sea surface temperature simulations) is represented by the second bar; and the surface-temperature-induced response (slow response; estimated as the difference of the total response minus the fast response) is represented by the third bar. Uncertainty is quantified using the 90% confidence interval; unfilled bars denote responses that are not significant at the 90% confidence level. Units are W m$^{-2}$.

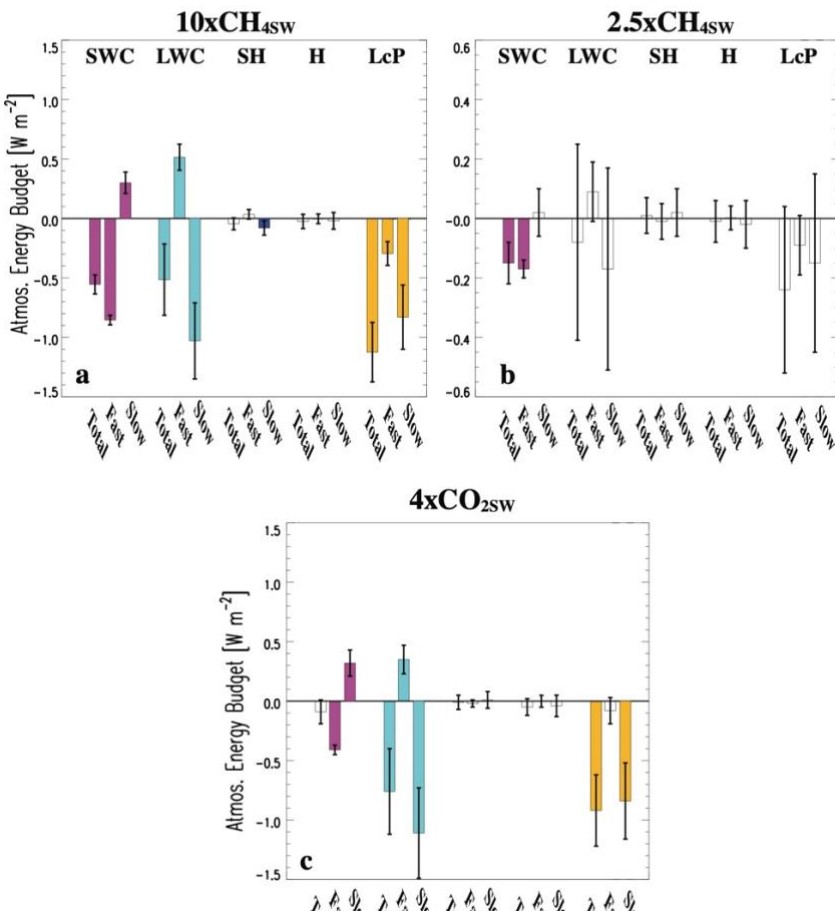

1425

**Figure 5. Atmospheric energy budget decomposition for the total, fast and slow response.** Annual mean global mean energy budget decomposition for (a) 10xCH4SW; (b) 2.5xCH4SW and (c) 4xCO2SW. Components include net shortwave radiative cooling from the atmospheric column (SWC); net longwave radiative cooling from the atmospheric column (LWC); net downwards sensible heat flux at the surface (SH); and column integrated dry static energy flux divergence (H). Positive values indicate cooling (energy loss). Also included is total latent heating ($L_cP$). The sum of the first four terms is equal to the last term ($L_cP$). The total response (from the coupled ocean atmosphere simulations) is represented by the first bar in each like-colored set of three bars; the rapid adjustment (fast response from fixed climatological sea surface temperature simulations) is represented by the second bar; and the surface-temperature-induced response (slow response; estimated as the difference of the total response minus the fast response) is represented by the third bar. Uncertainty is quantified using the 90% confidence interval; unfilled bars denote responses that are not significant at the 90% confidence level. Units are W m$^{-2}$. Note the different y-axis in panel b.

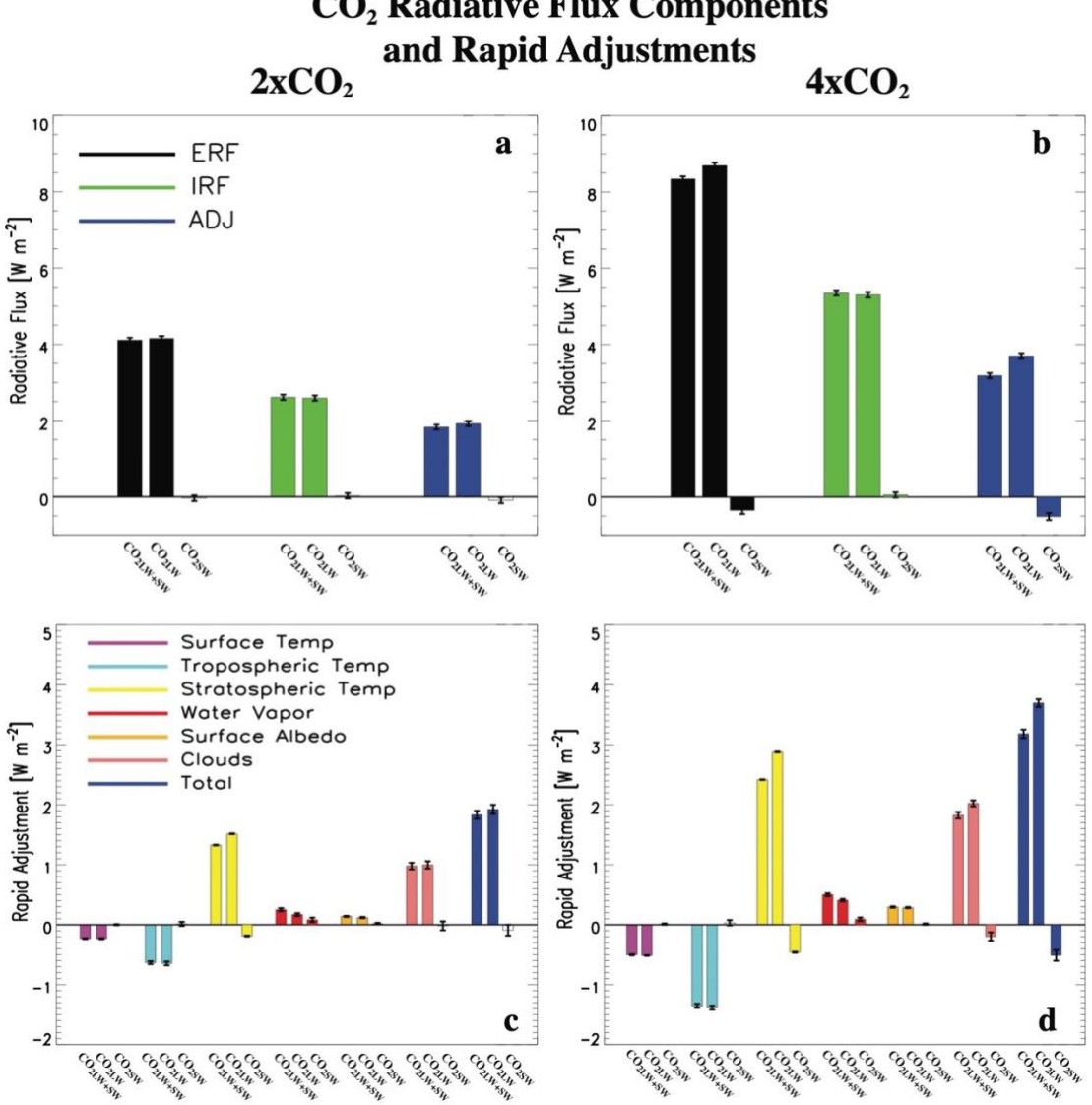

1442

**Figure 6. 2xCO$_2$ and 4xCO$_2$ top-of-the-atmosphere radiative flux components and rapid adjustments.** Global annual mean TOA (a, b) effective radiative forcing (ERF; black), instantaneous radiative forcing (IRF; green) and rapid adjustment (ADJ; blue); and (c, d) decomposition of the rapid adjustment into its components including surface temperature (purple), tropospheric temperature (cyan), stratospheric temperature (yellow), water vapor (red), surface albedo (orange), cloud (pink) and total rapid adjustment (blue) for (a, c) 2xCO$_2$ and (b, d) 4xCO$_2$. Responses are decomposed into CO$_2$ longwave and shortwave radiative effects (CO$_{2LW+SW}$), CO$_2$ longwave radiative effects (CO$_{2LW}$) and CO$_2$ shortwave radiative effects (CO$_{2SW}$). ERF and rapid adjustments are based on 30-year fixed climatological sea surface temperature simulations. Uncertainty is quantified using the 90% confidence interval; unfilled bars denote responses that are not significant at the 90% confidence level. Units are W m$^{-2}$.

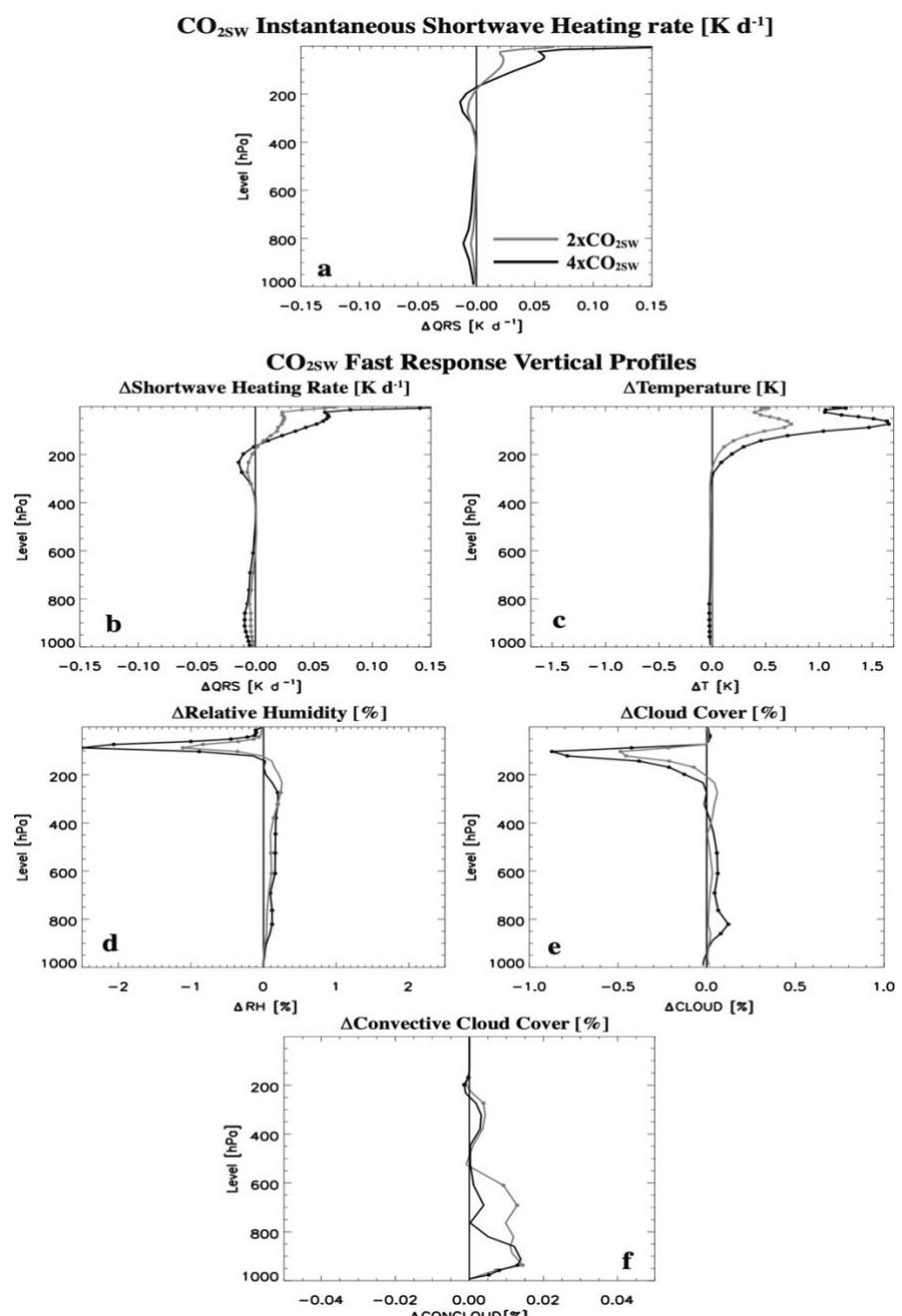

1456
**Figure 7. Global mean annual mean vertical response profiles for two CO$_{2SW}$**
**perturbations**. Instantaneous (a) shortwave heating rate (QRS; units are K d$^{-1}$);
and (b-f) fast responses of (b) QRS (units are K d$^{-1}$); (c) air temperature (T; units
are K); (d) relative humidity (RH; units are %); (e) cloud cover (CLOUD; units are
%) and (f) convective cloud cover (CONCLOUD; units are %) for 2xCO$_{2SW}$
(gray); and 4xCO$_{2SW}$ (black). A significant response at the 90% confidence level,
based on a standard t-test, is denoted by solid dots in (b-f). Climatologically fixed
SST simulations are used to estimate the fast responses. Instantaneous QRS
profiles come from the Parallel Offline Radiative Transfer Model (PORT).

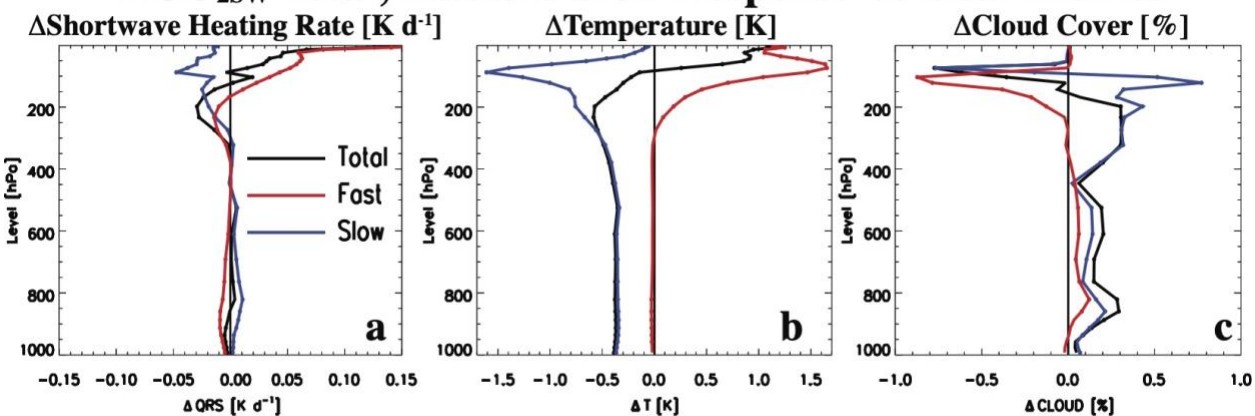

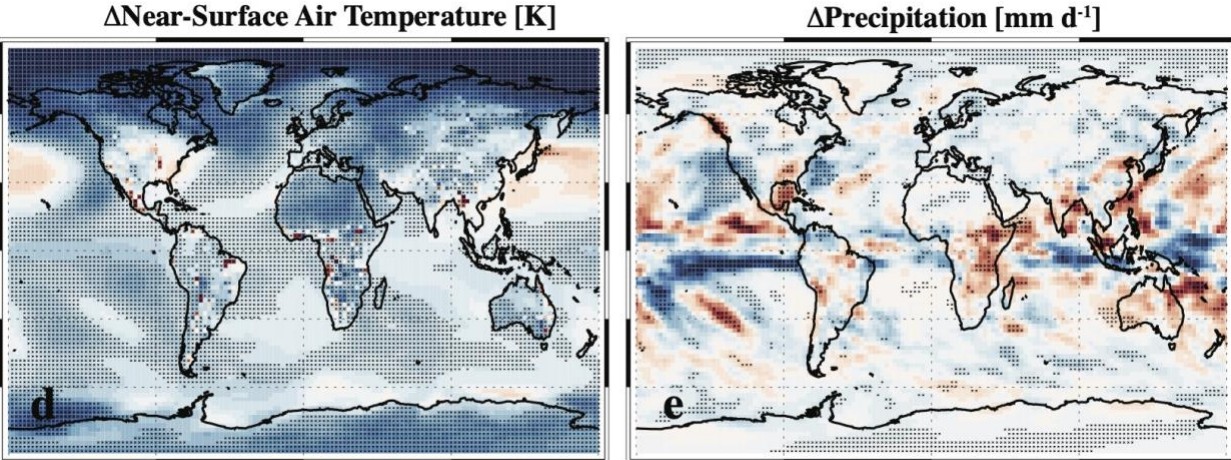

**Figure 8. 4xCO₂ₛᵥᵥ responses**. 4xCO₂ₛᵥᵥ annual mean global mean vertical response profiles of (a) shortwave heating rate (QRS; units are K d⁻¹); (b) air temperature (T; units are K); and (c) cloud cover (CLOUD; units are %) for the total (black); fast (red) and slow (blue) response. Also included are 4xCO₂ₛᵥᵥ global maps of the annual mean (d) near-surface air temperature [K] and (e) precipitation [mm d⁻¹] change for the total climate response. A significant response at the 90% confidence level, based on a standard t-test, is denoted by solid dots. Total climate responses are estimated using from coupled ocean-atmosphere CESM2 simulations.

**Feedback Decomposition**

**Figure 9. Feedback decomposition based on the radiative kernel method.**
Global annual mean top-of-the-atmosphere (TOA) surface temperature (purple),
tropospheric temperature (cyan), stratospheric temperature (yellow), water vapor
(red), surface albedo (orange), cloud (pink) and total (blue) feedback
decomposition, as estimated by normalizing the slow response's radiative flux
decomposition by the corresponding change in global mean near-surface air
temperature. Feedbacks are decomposed into $CH_4$ and $CO_2$ longwave and
shortwave radiative effects (e.g., $CH_{4LW+SW}$; first bar in each like-colored set of
three bars), longwave radiative effects (e.g., $CH_{4LW}$; second bar) and shortwave
radiative effects (e.g., $CH_{4SW}$; third bar). Uncertainty is quantified using the 90%
confidence interval; unfilled bars denote responses that are not significant at the
90% confidence level. Units are W m$^{-2}$ K$^{-1}$.