# Peer review of "Present-Day Methane Shortwave Absorption Mutes Surface Warming"

_EGUsphere, 2024_

## Author Comment (AC1)

**Response to Reviewer #1**

We thank Reviewer #1 for their comments and their evaluation of our paper. Below, we address each comment in blue.

**Reviewer #1**

Allen et al. assess the impact of methane (CH4) shortwave absorption for the increase of CH4 concentration from pre-industrial to present-day conditions.

The study builds on previous work (Allen et al., 2023) which has quantified the impact of CH4 shortwave absorption for idealized CH4 perturbations (2x, 5x, 10x pre-industrial CH4). The present study extends the analysis by explicitly simulating the impact for the present-day CH4 concentration, which corresponds to an increase of 2.5x pre-industrial CH4. Consistent with the 2x, 5x, 10x CH4 experiments, the present study finds that shortwave absorption of methane significantly mutes the effect of its longwave absorption. The study extends the analysis by an assessment of the energy budget and by comparing the effect of methane shortwave absorption to the effect of CO2 shortwave absorption.

The results are presented in a clear and understandable way. In my opinion it is a useful contribution to the understanding of the role of methane shortwave absorption. And - considering methane's short atmospheric lifetime – the findings are further relevant for the scientific assessment of short-term climate change mitigation options.

Therefore, I recommend publication after some minor revisions detailed below.

We thank Review #1 for their comments and their evaluation of our paper.

**General comment**

In my opinion, the paper is clearly written throughout most of the text. However, there are some formulations that might be misleading, especially if used out of context. At some points, the formulations "negative ERF" or "surface cooling" "under SW absorption" are used. I understand that "SW absorption mutes/offsets the (total) ERF" or "the SW effect/contribution to the ERF is negative" is meant. However, especially the formulation "under SW absorption" might be misleading as it could also mean "total ERF/temperature response if SW absorption is accounted for". Therefore, I suggest to carefully review the formulations and adapt the text where it might be misleading.

Some examples are:

• l. 114: "For example, the global mean near-surface air temperature (TAS) response under 5xCH4SW and 10xCH4SW (Figure 1a) yielded significant global cooling at -0.23 and -0.39 K."

We have added additional text to avoid misinterpretation. Here, for example, we have added:

For example, the global mean near-surface air temperature (TAS) response under 5xCH$_4$sw and 10xCH$_4$sw (Figure 1a) yielded significant global cooling at -0.23 and -0.39 K.  We reiterate that this cooling is due to isolation of methane shortwave absorption alone; the total (including methane's longwave absorption) temperature response is significant warming at 0.45 and 0.85 K, respectively (i.e., longwave absorption effects dominate).

• l. 273: "This negative rapid radiative adjustment promotes a negative ERF under methane SW absorption. …"

We have added text to avoid misinterpretation: This negative rapid adjustment promotes a negative ERF under methane SW absorption (we reiterate that the negative ERF is due to isolation of methane shortwave absorption alone; methane's longwave effects still dominate the ERF).

• l. 297: ".., 2.5xCH4SW yields larger (10-20%) and more negative TOA and surface IRFs, ERFs, and ADJs. The larger negative ERFs (and ADJs) act to promote cooling."

I think that the SW contribution to TOA IRF (2.5CH4SW) is not even negative, but weakly positive (Fig. 2a)).

Yes, thank you for catching this mistake, as the TOA IRF here is positive (the surface IRF is negative).  We have fixed this: To summarize, relative to 2xCH$_4$sw, 2.5xCH$_4$sw yields a larger positive TOA SW IRF and a larger negative surface SW IRF, as well as larger (10-20%) negative TOA and surface ERFs and ADJs.  The larger negative ERFs (and ADJs) act to promote cooling.  We reiterate that these values are due to isolation of methane shortwave absorption alone.

• l. 644: The total rapid radiative adjustment for both CO2 perturbations is negative under SW radiative effects at …"

We have added: The total rapid adjustment for both CO$_2$ perturbations is negative under SW radiative effects at -0.06 W m$^{-2}$ for 2xCO$_2$ and -0.40 W m$^{-2}$ for 4xCO$_2$ (we reiterate that these negative values are due to isolation of CO$_2$ shortwave absorption alone; CO$_2$'s longwave effects still dominate the total rapid adjustment and ERF).

• l. 831: "… leading to a negative ERF.

We have added: CO$_2$ SW absorption yields qualitatively similar results to CH$_4$ SW absorption, including a negative ADJ that offsets the positive IRF, leading to a negative ERF (Fig. 7; we reiterate that these negative ADJ and ERF values are due to isolation of shortwave effects alone).

**Specific comments**

l. 80: The term "rapid adjustments" is used in the introduction without a detailed explanation, which follows in the Methods section. Please shortly explain the term in the introduction or refer to the Methods section.

We included a brief description of "rapid adjustments", i.e., surface temperature independent responses. We have added "See Section 2" here, which provides a more complete definition of the term "rapid adjustment".

l. 159: I assume that the simulations are all "time slice simulation" (=cyclic repetition of the boundary conditions every year). This is not explicitly stated.

We included "equilibrium simulations", but we now explicitly also include "time slice simulations (i.e., cyclic repetition of the imposed perturbation)".

l. 209: Here an explicit description how the surface temperature driven feedbacks (e.g. Fig. 5) are calculated is missing. I assume that they are also calculated using the kernel method, but with the climate variable from the coupled ocean experiments. The radiative effects of the slow response are then presumably calculated as difference between radiative effects of the fast and total response?

Yes, this is correct. We included this information in Section 3.4, "We apply the radiative kernel decomposition to the $2.5xCH_{4SW}$ coupled ocean-atmosphere simulation (Figure 5). The 'fast' responses from the fixed climatological SST runs (i.e., the rapid adjustments) and the surface-temperature-induced 'slow' climate feedbacks (i.e., the difference between the coupled ocean atmosphere and fixed climatological SST simulations) are also included".

We now include this information in the Methods section as recommended by the Reviewer: The total climate response, which includes the IRF, ADJs and the surface temperature response, is quantified using the coupled ocean-atmosphere experiments. Specifically, the radiative effects associated with the total climate response are estimated using the same radiative kernel decomposition as above, but applied to the coupled ocean-atmosphere simulation. The surface temperature responses (i.e., 'slow' response) are estimated as the difference between the coupled ocean atmosphere simulations and the climatologically fixed SST experiments. Similarly, the radiative effects associated with the slow response are calculated as the difference between the kernel-derived radiative effects of the total and fast responses.

Section 3.4 /Fig. 5:

• The second paragraph (l. 443-455) might be moved to section 3.1 as only the rapid adjustments are discussed.

We've decided to keep this paragraph here.

- The radiative effects of the total and slow response are not shown for CH4LW and CH4LW+SW. A figure similar to Fig. 2 b) could be added in the supplement as comparison

The focus here is on the CH$_4$sw radiative effects. Nonetheless, we have added this figure to the Supplement.

Section 3.5.:

I am a bit confused about the sign convention in this section, which makes it difficult to follow the discussion. Could you give more detail on how to calculate LWC and SWC? Do they represent the divergence of LW/SW radiative fluxes in the total atmospheric column (=loss or gain of radiative energy of the total atmospheric column)?

If yes, I would presume that SWC would lead to energy gain (=warming) for reference conditions as the net downward SW flux at TOA is larger than the net downward SW flux at the surface (see e.g. Fig. 7.2 in IPCC-AR6, The Physical Science Basis). The LWC should lead to energy loss (=cooling) for reference conditions as the net downward LW flux at TOA is more strongly negative than the net downward LW flux at the surface, is this correct? The combined effect of LWC and SWC would be cooling (=net energy loss) as the absolute value of LWC is larger than SWC.

Does a positive LWC / SWC represent cooling (= net energy loss) or warming (=net energy gain)?

We have clarified this in the revision: LWC is the net longwave radiative cooling of the atmosphere. SWC is the net shortwave radiative cooling of the atmosphere. The "C" stands for cooling, i.e., positive SWC and LWC represent cooling of the atmospheric column. In CESM2, positive longwave radiative fluxes are upwards, so LWC is calculated as the net LW radiation at the TOA minus that at the surface. In CESM2, positive shortwave radiative fluxes are downwards, so SWC is calculated as the net SW radiation at the surface minus the net SW radiation at the TOA (or equivalently, the negative of the net SW radiation at TOA minus that at the surface). Both terms are positive for cooling (energy loss). SH is the downwards sensible heat flux at the surface (i.e., positive values indicate atmospheric cooling). H is estimated as the residual between L$_c$P and Q. In the global mean, the circulation term (i.e., H) is zero, implying L$_c$P = Q. As Q is composed of LWC and SWC, this balance shows that condensational heating via precipitation is largely balanced by radiative cooling of the atmosphere.

l. 850: It might be worth mentioning here that chemical composition changes of O3 and stratospheric H2O also affect the temperature response and thereby the static stability in the upper troposphere and stratosphere (see e.g. Winterstein et al, 2019, their Fig. 8; https://doi.org/10.5194/acp-19-7151-2019; for the temperature response induced by O3 and H2O changes in the stratosphere). Could this affect the cloud adjustment processes?

Yes, thank you for making this point, which we have now added to the revision: We reiterate that our simulations do not include these methane indirect effects. Such effects not only impact the

ERF, but also the temperature response in the stratosphere and upper troposphere (Winterstein et al., 2019), which in turn may impact the cloud response.

**Typos / technical corrections:**

l. 74: Etminan et al., 2016 (misspelled)  Fixed.

l. 94 : "… isolate the effect of …"  Fixed.

l. 129: estimates (plural) Fixed.

l. 149: "targeted methane-only equilibrium climate simulations"- This implies simulations perturbed by methane only, but you also conducted CO2 experiments.  Yes, we have included a note that CO2 experiments are also performed.

l. 217: I assume it is 5% of ERF?  Yes, this has been added.

l. 325: Double mentioning of word correlation: "Correlations between … are significant." Fixed.

l. 352: Should the unit of static stability be "K/km"?  We calculate lower-tropospheric stability as the temperature difference between 600 hPa and 990 hPa, i.e., units of K.  This has been clarified in the revision.

l. 471: Avoid line breaking between - and corresponding number (-0.31 Wm-2). Fixed.

Captions of Supplementary Fig. 1, 4 and 5: "Total climate responses are estimated using from coupled ocean-atmosphere CESM2 simulations." – "using data from coupled ocean-atmosphere simulations"  Fixed.

Captions of Supplementary Fig. 2 and 3: "Annual mean global mean spatial fast responses": The data do not show the global mean, but the spatial distribution.  Fixed.

---

## Author Comment (AC2)

**Response to Reviewer #2 (William Collins)**

We thank Reviewer #2, William Collins, for his comments (several made us think a bit deeper about our results) and evaluation of our paper. Comments are addressed in blue.

This is a valuable paper that makes good progress in understanding how radiative adjustments and responses vary according to the vertical profiles of short-wave absorption. It will certainly be suitable for publication after addressing the issues below.

This study uses the terms "fast" and "slow" responses, and seems to implicitly link the two in suggesting that the slow response is in some sense an extension of the fast response (for instance in the first paragraph of section 3.3). This is a different framework to that of the 6th Assessment Report of IPCC Working Group I (AR6), which used the concepts of "adjustment" to an imposed forcing and a radiative "response" to a global mean temperature change (GSAT) e.g. their Box 7.1, Equation 7.1: Delta_N=Delta_F+alpha*Delta_T. An assumption of AR6 is that the "response" is almost entirely a function of GSAT, and is largely independent of the characteristics of the imposed forcing, whereas in this study there seems to be an implication that the response ("slow response" in this paper) does depend on the characteristics of the imposed forcing. The paper needs to explicitly acknowledge this difference in framing compared to the IPCC and state whether their results imply the IPCC assumption of radiative response being a function solely of GSAT is overly simplistic.

It's unclear where we suggest the slow response in an extension of the fast response. We do, however, suggest that the fast response significantly contributes to the total response (in the context of the shortwave signal). This is what the first paragraph in section 3.3 is stating: "Figure 4a-e shows global mean vertical total climate response profiles from the coupled ocean-atmosphere simulations for the four methane shortwave absorption perturbations (e.g., $2.5xCH_{4}SW$). The QRS, RH and CLOUD responses are similar to those from the fSST simulation (Fig. 3), which further highlights the importance of rapid adjustments to the total climate response."

To reiterate, our framework is to decompose the total response (directly estimated from coupled simulations) into a fast (surface temperature independent) response and a slow (surface temperature dependent) response:

Total Response = Fast Response + Slow Response

The fast response is directly estimated from the fSST simulations and includes the rapid adjustments. The slow response is estimated from the difference of the total and fast responses (i.e., coupled simulation minus fSST simulation). This is consistent with the IPCC framework, which uses the concepts of an adjustment to an imposed forcing (i.e., independent of surface temperature) and a radiative response to a global mean temperature change. It is also analogous to the methodology employed in other papers, including several PDRMIP papers (e.g., Samset et al., 2016; Myhre et al., 2017). We have made this point more explicit in the revision (under Methods).

To aid the above discussion it would be useful to also present the radiative responses in figures 5 and 10 as feedback terms by dividing by Delta_T, and compare CH4sw and CO2sw. Tables of Delta_N, Delta_F, Delta_T and alpha should also be presented (either in the main text or supplement) for all the experiments (CH4, CO2, LW+SW and NOSW) to help understand why the SW effect contributes different percentages for ERF and Delta_T.

We unfortunately no longer have the model output for years 1-50 (for the coupled simulations). These simulations were run some time ago, Cheyenne was decommissioned (where we ran the simulations), and we only transferred the data currently in use (e.g., years 51-90 for the coupled runs) to local storage. Thus, we are unable to estimate alpha using the standard technique (e.g., "Gregory" regression, as illustrated in Box 7.1). Although we created such plots based on years 51-90, they are not very useful given the lack of years 1-50 (which are important for estimation of alpha).

As such, we perform an alternate procedure to estimate alpha (given the data that we do have) from the slope of the ordinary least squares regression line that connects two points: ($\Delta$TAS, $\Delta$N) from the coupled simulations and ($\Delta$TAS, $\Delta$N) from the fSST simulations. We have added a Supplementary Table to the revision (as well as a Supplementary Figure), both of which are included below. We also estimate the climate feedback using the radiative kernel method.

**Supplementary Table R1**. Global mean top-of-the-atmosphere energy decomposition for $CH_4$ and $CO_2$ perturbations based on the equation $\Delta N=\Delta F+\alpha\Delta TAS$, where $\Delta N$ is the change in the global mean TOA net energy flux [W m$^{-2}$]; $\Delta TAS$ is the change in global mean near-surface air temperature [K]; $\Delta F$ is the change in the global mean TOA net energy flux [W m$^{-2}$] when $\Delta TAS$ = 0 (i.e., the effective radiative forcing, ERF); and $\alpha$ is the net feedback parameter [W m$^{-2}$ K$^{-1}$]. Here, $\Delta N$ and $\Delta TAS$ are calculated using 40 years (years 51-90) from the coupled ocean-atmosphere simulations. $\Delta F$ is approximated using 30 years (years 3-32) from atmosphere-only simulations which feature climatologically fixed SST and sea-ice distributions (fSST). Uncertainty is estimated as 1.65*square root of the pooled variance. The net feedback parameter $\alpha$ is calculated from the slope of the regression line that connects two points: ($\Delta TAS$, $\Delta N$) from the coupled simulations and ($\Delta TAS$, $\Delta N$) from the fSST simulations. Using the surface-temperature adjusted $\Delta N$ from fSST simulations yields similar results. Uncertainty in $\alpha$ is estimated as the 1-sigma uncertainty estimate of the slope (the regression accounts for uncertainty in both $\Delta TAS$ and $\Delta N$). Corresponding values for the climate sensitivity parameter ($\lambda$; K [W m$^{-2}$]$^{-1}$) are also included, obtained by regressing ($\Delta N$, $\Delta TAS$) from the coupled simulations and ($\Delta N$, $\Delta TAS$) from the fSST simulations. Also included is an alternate estimate of the climate feedback parameter [$\alpha_k$; W m$^{-2}$ K$^{-1}$] as estimated by normalizing the slow response's radiative flux decomposition (based on the radiative kernel method) by its corresponding change in global mean near-surface air temperature.

|  | 2.5xCH$_{4LW+SW}$ | 2.5xCH$_{4LW}$ | 2.5xCH$_{4SW}$ |
|---|---|---|---|
| $\Delta$N | $0.04 \pm 0.20$ | $0.21 \pm 0.20$ | $-0.17 \pm 0.31$ |
| $\Delta$F | $0.43 \pm 0.12$ | $0.53 \pm 0.10$ | $-0.10 \pm 0.13$ |
| $\Delta$TAS | $0.25 \pm 0.05$ | $0.36 \pm 0.05$ | $-0.10 \pm 0.07$ |

| | | | |
|---|---|---|---|
| $\alpha$ | $-1.70 \pm 1.39$ | $-1.00 \pm 0.86$ | $0.87 \pm 3.41$ |
| $\alpha_k$ | $-1.59 \pm 0.88$ | $-0.85 \pm 0.63$ | $1.03 \pm 3.29$ |
| $\lambda$ | $0.59 \pm 0.48$ | $1.00 \pm 0.86$ | $-1.14 \pm 4.46$ |

| | 5xCH$_{4LW+SW}$ | 5xCH$_{4LW}$ | 5xCH$_{4SW}$ |
|---|---|---|---|
| $\Delta N$ | $0.11 \pm 0.20$ | $0.24 \pm 0.20$ | $-0.13 \pm 0.28$ |
| $\Delta F$ | $0.98 \pm 0.12$ | $1.20 \pm 0.10$ | $-0.22 \pm 0.17$ |
| $\Delta TAS$ | $0.45 \pm 0.05$ | $0.68 \pm 0.05$ | $-0.23 \pm 0.07$ |
| $\alpha$ | $-2.21 \pm 0.88$ | $-1.56 \pm 0.49$ | $-0.40 \pm 1.60$ |
| $\alpha_k$ | $-2.14 \pm 0.53$ | $-1.52 \pm 0.29$ | $-0.40 \pm 1.30$ |
| $\lambda$ | $0.45 \pm 0.18$ | $0.64 \pm 0.20$ | $2.48 \pm 9.81$ |

| | 10xCH$_{4LW+SW}$ | 10xCH$_{4LW}$ | 10xCH$_{4SW}$ |
|---|---|---|---|
| $\Delta N$ | $0.33 \pm 0.20$ | $0.50 \pm 0.20$ | $-0.17 \pm 0.31$ |
| $\Delta F$ | $1.70 \pm 0.13$ | $2.14 \pm 0.08$ | $-0.44 \pm 0.15$ |
| $\Delta TAS$ | $0.85 \pm 0.05$ | $1.24 \pm 0.05$ | $-0.39 \pm 0.07$ |
| $\alpha$ | $-1.80 \pm 0.44$ | $-1.45 \pm 0.26$ | $-0.73 \pm 1.08$ |
| $\alpha_k$ | $-1.81 \pm 0.28$ | $-1.45 \pm 0.17$ | $-0.72 \pm 0.86$ |
| $\lambda$ | $0.55 \pm 0.13$ | $0.69 \pm 0.12$ | $1.37 \pm 2.02$ |

| | 4xCO$_{2LW+SW}$ | 4xCO$_{2LW}$ | 4xCO$_{2SW}$ |
|---|---|---|---|
| $\Delta N$ | $2.82 \pm 0.17$ | $3.06 \pm 0.20$ | $-0.24 \pm 0.26$ |
| $\Delta F$ | $8.46 \pm 0.12$ | $8.82 \pm 0.12$ | $-0.35 \pm 0.15$ |
| $\Delta TAS$ | $5.45 \pm 0.07$ | $5.84 \pm 0.08$ | $-0.38 \pm 0.12$ |
| $\alpha$ | $-1.18 \pm 0.06$ | $-1.11 \pm 0.06$ | $-0.31 \pm 0.93$ |
| $\alpha_k$ | $-1.33 \pm 0.05$ | $-1.30 \pm 0.06$ | $-0.91 \pm 0.68$ |
| $\lambda$ | $0.85 \pm 0.04$ | $0.90 \pm 0.05$ | $3.27 \pm 9.98$ |

[Figure]

**Supplementary Figure R1.** Global mean TOA energy decomposition for $CH_4$ and $CO_2$ perturbations based on the equation $\Delta N = \Delta F + \alpha \Delta TAS$, where $\Delta N$ is the change in the global mean TOA net energy flux [W m$^{-2}$]; $\Delta TAS$ is the change in global mean near-surface air temperature [K]; and $\Delta F$ is the change in the global mean TOA net energy flux [W m$^{-2}$] when $\Delta TAS = 0$ (i.e., the effective radiative forcing, ERF). Uncertainty is estimated as 1.65*square root of the pooled variance. $\alpha$ is the net feedback parameter [W m$^{-2}$ K$^{-1}$] and is calculated from the slope of the ordinary least squares regression line that connects two points: ($\Delta TAS$, $\Delta N$) from the coupled simulations and ($\Delta TAS$, $\Delta N$) from the fSST simulations. Uncertainty in $\alpha$ is estimated as the 1-sigma uncertainty estimate of the slope (the regression accounts for uncertainty in both $\Delta TAS$ and $\Delta N$).

Supplementary Table R1 (and Supplementary Figure R1) shows the climate feedback parameter is always larger (in magnitude) under the various SW+LW signals (e.g., 2.5xCH4$_{LW+SW}$) as

compared to the LW-only signal (e.g., 2.5xCH4LW), which suggests the climate system does not have to warm as much to offset the same TOA energy imbalance when SW effects are included. However, $\alpha$ has a relatively large uncertainty and it is not significantly different between the various SW+LW signals and the corresponding LW-only signals. For example, the climate feedback parameter is $-1.80 \pm 0.44$ W m$^{-2}$ K$^{-1}$ for 10xCH4LW+SW and $-1.45 \pm 0.26$ W m$^{-2}$ K$^{-1}$ for 10xCH4LW. The SW signal consistently (outside of 2.5xCH4SW) yields the smallest (negative) $\alpha$. The corresponding value for 10xCH4SW is $-0.73 \pm 1.08$ W m$^{-2}$ K$^{-1}$. Similar results are generally obtained using $\alpha_k$ (based on the kernels). We also note that the 2.5xCH4SW $\alpha$ has an unphysical positive value (but again with large uncertainty) at $0.87 \pm 3.41$ W m$^{-2}$ K$^{-1}$. Thus, the climate feedback parameter is not significantly different under the LW-only effects versus SW effects of $CO_2$ and $CH_4$. This uncertainty also helps to explain why the SW effect contributes different percentages (which are not significant) for ERF and $\Delta$TAS. Based on the global mean top-of-the-atmosphere energy decomposition equation ($\Delta N=\Delta F+\alpha\Delta TAS$), if $\Delta F$ is reduced by 20%, $\Delta$TAS should also be reduced by 20% assuming a constant $\alpha$. The aforementioned uncertainty in $\alpha$, however, contributes uncertainty to the percentage offsets.

Additional analyses (included in the revision), however, show that there are significant differences in the cloud feedback (largely due to low clouds) that lend additional support to the notion that the climate feedback parameter is different (less negative) under methane SW radiative effects.

In a similar vein, the atmospheric energy budget is often presented as L*Delta_P = k*Delta_T-Delta_F(atm)-Delta_SH (e.g. Thorpe and Andrews 2014, MacIntosh et al. 2016), i.e. separating out the Delta_T contribution to LWC+SWC. Where the response term k* Delta_T is again assumed to be independent of the characteristics of the imposed forcing. It is not quite clear from the text (section 3.5) whether the different forcing characteristics lead to different k*Delta_T responses. Is k constant? As with the TOA budget, it would be useful to have tables of Delta_P, Delta_T, Delta_F(atm), Delta_SH, and k for all experiments (CH4, CO2, LW+SW and NOSW) , not just those in figure 6. This would help understand the percentage contributions of SW and LW effects.

We follow a procedure similar to the above procedure, but for precipitation. The corresponding Table and Figure are included below (they have also been added to the Supplement of the revision).

**Supplementary Table R2**. Global mean precipitation decomposition for $CH_4$ and $CO_2$ perturbations based on the equation $L_c\Delta P=A+\eta\Delta TAS$, where $L_c$ is the latent heat of condensation of water vapor with a value of 29 W m$^{-2}$ (mm day$^{-1}$)$^{-1}$; $\Delta P$ is the change in the global mean precipitation [mm day$^{-1}$]; $\Delta TAS$ is the change in global mean near-surface air temperature [K]; A is an adjustment term that accounts for the change in precipitation independent of any change in surface temperature [W m$^{-2}$], which can be further decomposed into SWC+LWC+SH, where SWC is the net shortwave radiative cooling of the atmosphere; LWC is the net longwave radiative cooling of the atmosphere; and SH is the downwards sensible heat flux at the surface (positive values for these three terms indicate cooling and energy loss). The hydrological sensitivity parameter is $\eta$ [W m$^{-2}$ K$^{-1}$]. Here, $\Delta P$ and $\Delta TAS$ are calculated using 50

years (years 51-90) from the coupled ocean-atmosphere simulations. $\mathbb{A}$ is approximated using 30 years (years 3-32) from atmosphere-only simulations which feature climatologically fixed SST and sea-ice distributions (fSST). Uncertainty is estimated as 1.65*square root of the pooled variance. The hydrological sensitivity parameter $\eta$ is calculated from the slope on the ordinary least squares regression line that connects two points: ($\Delta$TAS, $\Delta$P) from the coupled simulations and ($\Delta$TAS, $\Delta$P) from the fSST simulations (i.e., see Supplementary Figure X2). Uncertainty in $\eta$ is estimated as the 1-sigma uncertainty estimate of the slope (the regression accounts for uncertainty in both $\Delta$TAS and $\Delta$P).

|  | $2.5xCH_{4LW+SW}$ | $2.5xCH_{4LW}$ | $2.5xCH_{4SW}$ |
|---|---|---|---|
| $L_c\Delta$P | $0.119 \pm 0.18$ | $0.355 \pm 0.18$ | $-0.235 \pm 0.28$ |
| A | $-0.303 \pm 0.09$ | $-0.215 \pm 0.07$ | $-0.088 \pm 0.09$ |
| SWC | $-0.204 \pm 0.03$ | $-0.030 \pm 0.02$ | $-0.174 \pm 0.03$ |
| LWC | $-0.151 \pm 0.09$ | $-0.245 \pm 0.07$ | $0.094 \pm 0.10$ |
| SH | $0.051 \pm 0.03$ | $0.061 \pm 0.02$ | $-0.009 \pm 0.03$ |
| $\Delta$TAS | $0.25 \pm 0.05$ | $0.36 \pm 0.05$ | $-0.10 \pm 0.07$ |
| $\eta$ | $1.83 \pm 0.87$ | $1.78 \pm 0.60$ | $1.84 \pm 3.68$ |

|  | $5xCH_{4LW+SW}$ | $5xCH_{4LW}$ | $5xCH_{4SW}$ |
|---|---|---|---|
| $L_c\Delta$P | $0.423 \pm 0.17$ | $1.023 \pm 0.16$ | $-0.60 \pm 0.22$ |
| A | $-0.649 \pm 0.08$ | $-0.509 \pm 0.08$ | $-0.140 \pm 0.12$ |
| SWC | $-0.492 \pm 0.03$ | $-0.048 \pm 0.03$ | $-0.445 \pm 0.04$ |
| LWC | $-0.275 \pm 0.09$ | $-0.566 \pm 0.08$ | $0.291 \pm 0.12$ |
| SH | $0.121 \pm 0.03$ | $0.103 \pm 0.02$ | $0.018 \pm 0.03$ |
| $\Delta$TAS | $0.45 \pm 0.05$ | $0.68 \pm 0.05$ | $-0.23 \pm 0.07$ |
| $\eta$ | $2.73 \pm 0.48$ | $2.49 \pm 0.29$ | $2.06 \pm 1.12$ |

|  | $10xCH_{4LW+SW}$ | $10xCH_{4LW}$ | $10xCH_{4SW}$ |
|---|---|---|---|
| $L_c\Delta$P | $0.72 \pm 0.16$ | $1.84 \pm 0.17$ | $-1.12 \pm 0.25$ |
| A | $-1.16 \pm 0.08$ | $-0.866 \pm 0.07$ | $-0.290 \pm 0.10$ |
| SWC | $-0.917 \pm 0.03$ | $-0.066 \pm 0.02$ | $-0.850 \pm 0.04$ |
| LWC | $-0.459 \pm 0.09$ | $-0.978 \pm 0.07$ | $0.519 \pm 0.11$ |
| SH | $0.220 \pm 0.03$ | $0.181 \pm 0.03$ | $0.039 \pm 0.04$ |
| $\Delta$TAS | $0.85 \pm 0.05$ | $1.24 \pm 0.05$ | $-0.39 \pm 0.07$ |
| $\eta$ | $2.47 \pm 0.24$ | $2.39 \pm 0.16$ | $2.24 \pm 0.73$ |

|  | $4xCO_{2LW+SW}$ | $4xCO_{2LW}$ | $4xCO_{2SW}$ |
|---|---|---|---|
| $L_c\Delta$P | $6.96 \pm 0.19$ | $7.87 \pm 0.20$ | $-0.913 \pm 0.30$ |
| A | $-4.91 \pm 0.09$ | $-4.84 \pm 0.07$ | $-0.077 \pm 0.11$ |
| SWC | $-0.579 \pm 0.03$ | $-0.173 \pm 0.03$ | $-0.406 \pm 0.04$ |
| LWC | $-3.99 \pm 0.09$ | $-4.34 \pm 0.08$ | $0.350 \pm 0.12$ |
| SH | $-0.368 \pm 0.03$ | $-0.353 \pm 0.02$ | $-0.016 \pm 0.03$ |

| ΔTAS | $5.45 \pm 0.07$ | $5.84 \pm 0.08$ | $-0.38 \pm 0.12$ |
|---|---|---|---|
| η | $2.47 \pm 0.04$ | $2.46 \pm 0.04$ | $2.31 \pm 0.89$ |

[Figure]

**Supplementary Figure R2.** Global mean precipitation decomposition for $CH_4$ and $CO_2$ perturbations based on the equation $L_c\Delta P = A + \eta\Delta TAS$, where $L_c$ is the latent heat of condensation of water vapor with a value of 29 W m$^{-2}$ (mm day$^{-1}$)$^{-1}$; $\Delta P$ is the change in the global mean precipitation [mm day$^{-1}$]; $\Delta TAS$ is the change in global mean near-surface air temperature [K]; A is an adjustment term that accounts for the change in precipitation independent of any change in surface temperature [W m$^{-2}$]. Uncertainty is estimated as 1.65* square root of the pooled variance. η is the hydrological sensitivity parameter [W m$^{-2}$ K$^{-1}$] and is calculated from the slope of the ordinary least squares regression line that connects two points:

(ΔTAS, ΔP) from the coupled simulations and (ΔTAS, ΔP) from the fSST simulations. Uncertainty in η is estimated as the 1-sigma uncertainty estimate of the slope (the regression accounts for uncertainty in both ΔTAS and ΔP).

Supplementary Table R2 (and Supplementary Figure R2) shows the hydrological sensitivity parameter is always larger (in magnitude) under the various SW+LW signals (e.g., 2.5xCH$_{4LW+SW}$) as compared to the LW-only signal (e.g., 2.5xCH$_{4LW}$). The SW signal consistently (outside of 2.5xCH$_{4SW}$) yields the smallest η. However, η has a relatively large uncertainty and it is not significantly different between the various SW+LW signals and the corresponding LW-only and SW signals. For example, the hydrological sensitivity parameter is $2.47 \pm 0.24$ W m$^{-2}$ K$^{-1}$ for 10xCH$_{4LW+SW}$ and $2.39 \pm 0.16$ W m$^{-2}$ K$^{-1}$ for 10xCH$_{4LW}$. The corresponding value for 10xCH$_{4SW}$ is $2.24 \pm 0.73$ W m$^{-2}$ K$^{-1}$. Thus, the hydrological sensitivity parameter is not significantly different under the LW-only effects versus SW effects of CO$_2$ and CH$_4$. These points have been added to the revision.

The term "feedback" is conventionally (e.g. AR6) used to describe the radiative response to a 1 K change in GSAT in W/m2/K, whereas in this paper it is used as a synonym for the radiative response itself in W/m2. This could lead to confusion for readers, for instance section 3.4 refers to "negative feedback" several times for processes that amplify the original signal and would conventionally be described as "positive feedbacks".

We have removed the use of the word "feedback" throughout Section 3.4 and in other similar places. Instead, we use "slow response". We note that we now add an explicit feedback analysis based on the slow response (e.g., the corresponding radiative fluxes from the kernels are normalized by DTAS).

Figure 5 shows that the TOA radiative imbalance increases once the ocean is allowed to respond. This is contrary to expectations that the radiative response to the temperature change should be such that it brings the system closer to radiative balance until it reaches equilibrium. If the TOA imbalance is increasing this will imply that the climate system will continue to cool indefinitely without reaching equilibrium. The overall climate feedback term (which should be calculated) is a negative flux change divided by a negative temperature change and is therefore positive and would lead to a snowball Earth which seems unphysical. This concern needs to be addressed by the authors. This may come about because the 'SW' simulation is not carried out explicitly, but as the difference between the LW+SW and NOSW simulations. The time evolution of the net TOA imbalance over the 90 years of the simulations needs to be plotted to understand what is happening.

Yes, Figure 5 shows that the total radiative flux (sum over clouds, water vapor, etc.) for the slow response under 2.5xCH$_{4SW}$ is negative (opposite expectations). However, there is large uncertainty, i.e., it is a nonsignificant negative value at $-0.10 \pm 0.30$ W m$^{-2}$. Uncertainties have been added in this figure and throughout the revision. We note that over all of our experiments, this is the only instance where this occurs.

To go into more details, the total radiative flux decomposition for the total response under 2.5xCH$_{4SW}$ (based on coupled simulations) is more negative than the rapid adjustment (from

fixed climatological SST experiments) at $-0.27 \pm 0.28$ W m$^{-2}$ and $-0.16 \pm 0.10$ W m$^{-2}$, respectively. The former number (-0.27) is based on the total radiative flux decomposition under 2.5xCH$_{4SW+LW}$ minus 2.5xCH$_{LW}$, which have respective values of $-0.46 \pm 0.18$ W m$^{-2}$ and $-0.19 \pm 0.19$ W m$^{-2}$. So here, both values are negative, as expected (i.e., the climate system responds to the positive forcing by warming and emitting more energy to space).

Looking at it another way, the 2.5xCH$_{4SW}$ ERF is $-0.10 \pm 0.13$ W m$^{-2}$. The corresponding net energy imbalance ($\Delta N$) from the coupled simulations (averaged over the last 40 years of the 90 year integration) is more negative (but again with large uncertainty) at $-0.17 \pm 0.31$ W m$^{-2}$ (e.g., Supplementary Table R1), which again shows the unexpected signal. The $-0.17 \pm 0.31$ W m$^{-2}$ value comes from the $\Delta N$ difference between 2.5xCH$_{4SW+LW}$ and 2.5xCH$_{4LW}$, which have values of $0.04 \pm 0.20$ W m$^{-2}$ and $0.21 \pm 0.20$ W m$^{-2}$, respectively. The latter number is perhaps larger than expected (which leads to the relatively large $-0.17 \pm 0.31$ W m$^{-2}$). The 2.5xCH$_{4LW}$ ERF is $0.53 \pm 0.10$ W m$^{-2}$, which means the $\Delta N$ from the 2.5xCH$_{4LW}$ coupled simulation has acted to offset only 60% of the ERF after 90 years, which is a lower percentage than the other experiments (e.g., 80% for 5xCH4).

Consistent with the next comment, longer integrations of our experiments would have been ideal. Given computational resource limitations, there will always be a tradeoff between the number of simulations performed and length of each simulation. With 4 methane perturbations and a carbon dioxide perturbation for LW+SW and NOSW, plus 3 corresponding control simulations, we have performed about 1200 simulations years (not counting the fSST runs, which adds nearly another 500 simulation years).

As mentioned above, we no longer have restart files so it is unfortunately not possible to extend any of the simulations.

The 90 year simulations will not be long enough for the full climate response to manifest itself (they obviously have not reached radiative balance). This should be discussed. It is likely that with a coupled ocean the interannual variability will lead to high levels of uncertainty in the calculations, even with a 40-year average. A longer average or a number of ensemble simulations might be needed to reduce this. Uncertainties should be quoted on all values and should be presented on the bar graphs.

Yes, we agree (sorry for inadvertently omitting). Please see the prior comments. We have added this point to the revision (e.g., Methods).

Uncertainties have been added.

It is not obvious that it is necessary to discuss in such detail the comparison between 2x and 2.5x CH4. I understand that part of the motivation for this study was to understand why the 2x CH4 behaved so differently in A23 compared to the larger perturbations. However, for a reader that hadn't been familiar with the A23 discussions the comparisons between 2x and 2.5x are likely to distract from the main messages. Could this be moved to the supplement?

We have removed much of the 2xCH$_4$ and 2.5xCH$_4$ comparison.

The qualitive comparisons between CH4sw and CO2sw (particularly the height profiles) are very informative, but the quantitative discussions of percentage offsets are less useful. The SW component of the forcing is much larger for CH4 than for CO2 so obviously it will have a much larger percentage offset irrespective of the difference in height profiles.

We have toned down some of the quantitative discussion here.

Line 2: From section 3.3 the precipitation effect isn't significant.

We have removed "wetting" from the title.

Line 24: This sentence needs to be rephrased to reflect that the precipitation effect isn't significant (line 407).

Rephrased.

Lines 43-45: This sentence needs to be rephrased to reflect that the precipitation effect isn't significant (line 407).

Rephrased to indicate the precipitation effect is not significant.

Lines 48-52: These two sentences seem to imply that the difference between methane and carbon dioxide SW effects is mostly due to the vertical profile, whereas the main difference is simply that carbon dioxide absorbs less strongly in the SW compared to the LW.

We have re-ordered these two sentences.

Lines 132-135: If the linear fits go through zero, then the percentage offsets should be the same at any point on the line. By zooming in repeatedly on figure 1(a, b) I was able to see that the lines don't go through zero. This can't make sense since zero change in CH4 must give zero change in Delta_T or Delta_P. Hence this disagreement is purely an artifact of the fitting. Note that in line 418 there is a suggestion of a logarithmic relationship, which would mean that the linear fits in figure 1(a, b) aren't appropriate anyway.

We note that the regression line was fit by including the (0, 0) data point (i.e., $\Delta$TAS and $\Delta$P are zero when the change in methane concentration is zero).  However, we did not constrain the y-intercept of the regression to be zero.  Our intention was to reproduce the methodology from A23, and to show the difference between the estimated values of $\Delta$TAS and $\Delta$P for the present-day methane concentration (from A23) versus what we do here (i.e., explicit simulation).

Nonetheless, based on this and related comments, we have toned down our comparison of the 2xCH4 simulations from A23 to the new 2.5xCH4 simulations emphasized here.  We have removed this figure from the revision.  We have also removed the discussion near L418.

Lines 223-226: There needs to be some justification provided as to whether these time periods are sufficient.

We have added some information here:  Our integration lengths are consistent with other related idealized time-slice studies including for example a 100-year integration (and analysis of the last 50 years) of coupled simulations under PDRMIP (e.g., Samset et al., 2016; Myhre et al., 2017). A similar statement applies for the integration length of our fSST runs, e.g., the Radiative Forcing Model Intercomparison Project (RFMIP; Pincus et al., 2016) specifies 30-year fSST simulations.

We note that even with a 90-year coupled ocean simulation, the model has not yet reached equilibrium.  Given computational resource limitations, there is always a tradeoff between the number of simulations performed and length of each simulation.

NOTE: We'd extend these runs if we could, but we no longer have the restart files as Cheyenne was decommissioned (see additional related comments below).

Page 7-8: The comparisons to 2xCH4 just add a lot of extra values and don't add any extra science. The 2xCH4 values from the text, and fig 2(c,d)  could be provided in the supplement for anyone who wanted to look them up.

Much of this has been removed.

Lines 304-324: Comparisons with 2xCH4 aren't needed.

Removed.

Line 353: Explain how this lower-tropospheric stability is defined/quantified.

We have added this information: Global mean lower-tropospheric stability is estimated here as the temperature difference between 600 hPa and 990 hPa.

Line 355-356: Would it be better to say that "the increase in low cloud cover is consistent with the increase in lower-tropospheric stability" rather than the other way around?

Change incorporated.

Section 3.3: I think this would be much clearer to understand if the response to surface temperature ("slow" in the notation of this study) were isolated from the total response. Showing just the sum of the adjustment and the response risks conflating the two and implying that they have a common cause. If the authors do wish to make the point that the adjustments and responses are more closely linked than in the IPCC AR6 framework they should make the point explicitly.

The slow (temperature-induced) response is isolated from the total response as the difference: slow response = total response from coupled simulations minus fast response/adjustment from fSST simulations.  The slow response is presented in the subsequent section (3.4).  We reiterate that we are making the point that the total and fast responses have similarities.  We are not trying to make the point that the slow and fast responses have similarities.  We have clarified our

decomposition in the revision and we have also clarified by improving our terminology (as recommended in this review, i.e., replacing "feedback" with slow response, etc.). We also note that we have added an explicit feedback analysis.

Lines 386-389: It doesn't really add scientific value to compare the magnitude of the lower tropospheric temperature adjustment to the total response since the first is strongly constrained by the fixed SST.

Yes, we agree that the tropospheric temperature adjustment will be much smaller than that under the total response, since the adjustment comes from fSST simulations. This paragraph discusses multiple adjustments, so we retain the tropospheric temperature adjustment for completeness. Nonetheless, we have added a sentence that indicates that this is expected, since the adjustment is based on fSST simulations.

Lines 398-399: The uses of "muted" and "augmented" imply again that the adjustments and responses are more closely linked than in the IPCC AR6 framework. If this is the intention this point should be made explicitly.

Please see above comments. We are comparing the total response to the adjustment (fast response). Our main point, in the context of the ΔCLOUD profiles, is that the total CLOUD response is similar to (but larger than) the fast CLOUD response.

Line 400: An example of the possible confusion by presenting the total rather than separating out the response to temperature comes from the sentence "The total response of "CONCLOUD is generally similar to the fast response …" At first reading I took this to be that that the temperature-driven response is similar to the adjustment, then I realised that for the total to be equal to the adjustment, then the response has to be zero.

We are trying to show the total response shares similarities to the fast response/adjustment.

Lines 408-410: It appears that the ratio of temperature change to ERF is much larger (-0.1/-0.1) for the SW than the LW (0.35/0.53). And this is the reason a larger fraction of the warming (29%) is offset compared to the forcing (19%). This should at least be commented on. Does it appear that the climate system has a greater climate sensitivity to the pattern of forcing from the SW? How robust is this? Using an uncertainty in the temperature of +-0.04K (from line 42, but should be quoted in this section), the ERF and warming fractions are consistent within uncertainties.

Please see above. We have added analyses that address the climate feedback parameter.

Line 411: Presumably this 66% is not significant if the change in precipitation is not significant.

This is correct—the change in precipitation under $2.5xCH_{4SW}$ is not significance at the 90% confidence level. This was explicitly stated in the original submission, but has been stated more firmly throughout the revision, in accord with this comment and several prior comments. As such, the quoted 66% offset is also not significant. This has also been clarified in the revision.

We reiterate that one of the main goals of this paper was to clarify the 2xCH4sw signal from A23, by performing the 2.5xCH4 simulations emphasized in this paper. Recall, the 2xCH4sw signal yielded opposite (but not significant) global mean changes in near-surface air temperature (warming of $0.06 \pm 0.06$ K) and precipitation (increases of $0.002 \pm 0.008$ mm d$^{-1}$) as compared to the larger methane perturbations. The 2.5xCH4sw signal, however, yields changes in better agreement with those from the larger methane perturbations from A23, including muted warming and muted wetting (the point we tried to make in the original submission). It is true, however, that some of these 2.5xCH4sw signals remain insignificant at the 90% confidence level, such as the precipitation signal discussed here ($-0.008 \pm 0.009$ mm d$^{-1}$). We understand the concern and the importance of emphasizing significant versus insignificant signals, and we have improved this distinction in the revision. But we also point out that the 2.5xCH4sw signal is in better agreement to the larger methane perturbations from A23.

It is obviously more difficult to quantify the 2.5xCH4sw signals, due to the weaker perturbation relative to internal climate variability. But the consistency of the 2.5xCH4sw signals relative to those under the larger methane perturbations (5xCH4sw and 10xCH4sw) supports the robustness of the main conclusions regarding the importance of methane SW absorption.

Lines 412-417: I'm not sure the discussion of the warming patches is useful since they are unlikely to be robust.

These lines have been deleted.

Lines 418-428: The methane forcing is typically assumed to vary with the square root of the concentration (e.g. Etminan et al. 2016). This paragraph asserts that the variation is logarithmic. Are the uncertainties small enough to discount the square root dependence? Is there a physical reason why it should be logarithmic? Are the SW bands nearer saturation than the LW ones?

Now that we have the 2.5xCH4sw simulations, the intention here was to show consistent/proportional changes between the change in atmospheric methane concentration and the climate signal (e.g., $\Delta$TAS). Here, this consistency was illustrated by noting that every doubling of the atmospheric methane concentration yields a doubling of $\Delta$TAS. However, given the uncertainty in the signals (which we have now thoroughly emphasized in this revision), it is likely not possible to discriminate the logarithmic versus square root dependency. This has been deleted.

Line 458: The total feedback is positive (which is in itself worrying) and these terms make a positive contribution to the feedback (since they are negative radiative responses to a negative temperature change).

Yes, please see the comment above.

Line 462: In IPCC terminology these are "responses" not "adjustments", again they are positive responses that contribute a negative feedback to an overall positive total feedback.

Yes, this was an inconsistent use of our terminology. We have changed to "slow responses".

Lines 464-490: This paragraph compares the "fast" and "slow" responses, but under the IPCC framework there is no link between "adjustment" and "response" (apart from through global mean temperature). This leads to some confusing statements. For instance "… lack of an increase in upper-tropospheric heating rate" makes it sound as if something is missing in the response, but since the mechanisms are completely different there is no connection between the impact of an imposed forcing and the response to a global mean surface temperature change.

 I think in all cases "feedback" needs to be replace with "response". This needs to be checked.

Changed.

Lines 492-494: is the difference between 19% and 29% significant? The 66% precipitation increase is not significant.

We have clarified.  The 29% muting of the surface warming is significant at the 90% confidence level; the other two are not.  As mentioned above, the 2.5xCH$_{4SW}$ precipitation change is not significant at the 90% confidence level at $-0.008 \pm 0.009$ mm d$^{-1}$.  The corresponding ERF also lacks significance at the 90% confidence interval at $-0.10 \pm 0.13$ W m$^{-2}$.

Lines 503-549: In order to follow the discussion in these paragraphs, it would be very useful to have a table of all the numbers for figure 6, with the sum of the components (which presumably should be exactly equal to LcP if energy is conserved in the model). Does the hydrological sensitivity of the "slow" mode here agree with e.g. Flaschner et al. 2016 of ~2.2 W/m/2/K?

These numbers from Figure 6 are now included in a Table (e.g., Supplementary Table R2 above).  Yes, the sum of the components is equal to LcP.  For example, under 2.5xCH$_{4SW}$, SWC is $-0.174 \pm 0.03$ W m$^{-2}$; LWC is $0.094 \pm 0.10$ W m$^{-2}$; and SH is $-0.009 \pm 0.03$ W m$^{-2}$.  These three terms sum to $-0.089$ W m$^{-2}$.  The corresponding LcP calculated directly from precipitation is $-0.088 \pm 0.09$ W m$^{-2}$.  The hydrological sensitivity $\eta$ is also included in Supplementary Table R2 above.  $\eta$ under 2.5xCH$_{4SW}$ is $1.84 \pm 3.68$ W m$^{-2}$ K$^{-1}$; corresponding values under 5xCH$_{4SW}$, 10xCH$_{4SW}$ and 4xCO$_{2SW}$ are $2.06 \pm 1.12$ W m$^{-2}$ K$^{-1}$; $2.24 \pm 0.73$ W m$^{-2}$ K$^{-1}$ and $2.31 \pm 0.89$ W m$^{-2}$ K$^{-1}$, respectively.  Uncertainty is relatively large (especially under 2.5xCH$_{4SW}$), but all of these values are consistent with a hydrological sensitivity of ~2.2 W m$^{-2}$ K$^{-1}$.  We also note that given the uncertainty, $\eta$ under LW+SW versus LW-only versus SW-only effects are not significantly different.  For example, $\eta$ under 2.5xCH$_{4LW+SW}$ is $1.83 \pm 0.87$ W m$^{-2}$ K$^{-1}$; $\eta$ under 2.5xCH$_{4LW}$ is $1.78 \pm 0.60$ W m$^{-2}$ K$^{-1}$; and $\eta$ under 2.5xCH$_{4SW}$ is $1.84 \pm 3.68$ W m$^{-2}$ K$^{-1}$.

Lines 560-569: It should be made clear here that the difference in behaviour between BC and CH4sw is that BC has a positive TOA ERF whereas CH4sw has a negative TOA ERF. However both species have a positive atmospheric ERF.

We have added this clarification to the revision.

Lines 575-582: I find this discussion of the QRS profile rather confusing. It is the total atmospheric forcing (i.e. difference between TOA and surface) that matters, the vertical

distribution is not important. It may be the authors are trying to make the point that the atmospheric forcing from CH4sw has positive and negative regions that the total forcing is less than if it were uniformly positive like BC.

We have removed this discussion.

Lines 583-585: It is not made very clear here why the different vertical heating profiles of CH4sw and BC should lead to different ERFs. I think the argument from section 3.3 is that because the QRS for CH4sw is negative in the lower troposphere the low cloud adjustment is sufficiently negative to make the total ERF negative, whereas because the QRS for BC is positive in the lower troposphere there isn't such a negative low cloud adjustments and the overall ERF is positive. If this is indeed the argument, it should be stated more explicitly.

Yes, we are suggesting that differences in the vertical QRS profile lead to different clouds adjustments, which in turn impact the ERF. For example, in a prior paper (https://www.nature.com/articles/s41612-019-0073-9), we show that how QRS is vertically distributed in the atmosphere has large impacts on the cloud adjustment and in turn the ERF. Although that paper is not exactly analogous to the situation here, we found that vertically uniform heating leads to increases in low clouds and decreases in high clouds, and in turn, a negative cloud adjustment (mostly due to LW effects) and a negative ERF (e.g., Table 1 from the above paper). In contrast, putting the same heating essentially in the lower troposphere leads to decreases in low clouds and weak decreases in high clouds, a positive cloud adjustment (mostly due to SW effects) and a positive ERF. Since this is more speculation (we have not performed any BC experiments), we have moved this idea to the discussion section.

Section 3.6: Uncertainties need to be provided on all values here in order to be able to work out whether similarities or differences with CH4 are significant.

Uncertainties added.

Lines 714: I don't quite see how the vertical temperature profile is not consistent with the QRS profile for the total climate response.

We've deleted these lines.

Lines 725: It is interesting that the temperature responses to CO2sw and CH4sw are similar (~ 1K per W/m2), and are higher than for the CH4lw (0.7K per W/m2). The response for CO2lw should be provided as well. Are these increased climate sensitivities for SW significant? Is there any physical explanation?

We have added a Table (included above) that includes an analysis of the climate feedback parameter, $\alpha$. As previously discussed, the climate feedback parameter is always more negative under the various SW+LW signals (e.g., 2.5xCH4$_{LW+SW}$) as compared to the LW-only signal (e.g., 2.5xCH4$_{LW}$), which suggests the climate system does not have to warm as much to offset the same TOA energy imbalance when SW effects are included. The SW signal consistently (outside of 2.5xCH4$_{SW}$) yields the smallest (negative) $\alpha$. However, $\alpha$ has relatively large

uncertainty and it is not significantly different between the various signals. Analogous conclusions exist for the climate sensitivity parameter $\lambda$ (K [W m$^{-2}$]$^{-1}$; i.e., $-1 \times \alpha^{-1}$ ). $\lambda$ is consistently smaller under the various SW+LW signals relative to the corresponding LW-only signals, implying less warming in response to the same TOA energy imbalance when SW effects are included.  The SW signal (outside of 2.5xCH4sw) consistently yields a larger $\lambda$, implying relatively large temperature changes in response to the same TOA energy imbalance.  Again, however, the uncertainty is large and these differences are not significant.  For example, the climate sensitivity parameter is $0.55 \pm 0.13$ K [W m$^{-2}$]$^{-1}$ under 10xCH4LW+SW versus $0.69 \pm 0.12$ K [W m$^{-2}$]$^{-1}$ under 10xCH4LW.  The corresponding $\lambda$ under 10xCH4SW is $1.37 \pm 2.02$ K [W m$^{-2}$]$^{-1}$.

Lines 732-735: Again, it is not quite clear whether this is trying to suggest that the slow cloud response is an amplifying effect particular to the SW forcing (and is similar for CH4 and CO2), rather than the slow cloud response being common to any global mean temperature change, and in these cases happens to add to the adjustment.

Our intent was the latter.  We've deleted this line.

Lines 736-748: I'm not convinced this discussion comparing the SW and LW effects adds anything useful. CO2 has a much smaller SW component.

We've decided to retain this paragraph.  But we agree with the reviewer; this paragraph is another way of saying CO2 has a much smaller SW component.

Lines 782-797: I appreciated the authors wanted to understand their A23 results, but this discussion of simulated vs inferred is likely to be of little interest to most readers. It should be moved to the supplement along with figures 1(c-e).

This discussion has been removed.

Lines 816-820: For the fast responses it is only the integral of the QRS profile that matters, not whether it is positive or negative at particular levels. The difference in behaviour between CH4sw and BC comes from the different signs of the global temperature response which is driven by the ERF. I think the point is that the negative QRS in the lower troposphere leads to a negative low cloud adjustment for CH4sw and hence negative ERF, whereas for BC the positive QRS in the lower troposphere leads to less low cloud adjustment so the ERF is overall positive. This should be clarified here.

We have clarified.

Lines 829-846: I don't think the quantitative comparison of the various percentages of offsets between CH4sw and CO2sw is a key conclusion since CO2sw is only a small component of the CO2 forcing.  Rather, the qualitative understanding of the impacts of different QRS profiles on adjustments and precipitation for CH4sw, CO2sw and BC is an important finding and should be expanded on explicitly here.

We have removed much of this quantitative comparison.

We have added some additional discussion on the vertical QRS profiles and precipitation changes.

Lines 864-871: Note that the IPCC AR6 CH4sw values are based on 4 models, one of which is CESM1. Therefore care needs to be taken when comparing with IPCC not to compare the model with itself.

We have added a note that the IPCC's estimate is based on a similar model used here.

Line 817: The IPCC AR6 estimates an overall increase in methane ERF from including SW absorption. There is a +0.12 W/m2 increase in SARF (from Etminan et al. 2016 and largely due to including SW absorption) and a -0.08 W/m2 decrease due to cloud adjustments from Smith et al. 2018, leaving an overall increase of +0.02 in ERF from including SW. Hence I think this study would lower the methane radiative efficiency. The impact of the adjustments on the Etminan radiative efficiency could be estimated here by adding the adjustments from figure 2, assuming that the Etminan value only includes stratospheric temperature adjustment.

We're not sure about this. Should the 0.02 overall increase in ERF be 0.04 W/m^2? Our 2.5xCH4sw total rapid adjustment minus the stratospheric adjustment minus the cloud adjustment is -0.16-(-0.04)-(-0.12) = 0.0 W/m^2 (i.e., the other terms have net zero contribution).

Lines 878-900: I don't think the emission-based discussion is useful here as it detracts from the key messages on adjustments.

We've retained this paragraph. Although our key message pertains to CH4sw adjustments, we feel it is also necessary to (briefly) discuss additional aspects of methane.

Line 902: The uncertainty on this 30% needs to be added. The smaller decrease (19%) in ERF needs mentioning too, since it is not obvious why this is amplified in the temperature, and whether this amplification is robust.

Uncertainty added. As addressed above (and included in the revision), this amplification of temperature relative to ERF does not appear to be robust, given the relatively large uncertainty in the climate feedback parameter.